# Detecting and Mitigating Memorization in Diffusion Models through Anisotropy of the Log-Probability

**Rohan Asthana, Vasileios Belagiannis**
Friedrich-Alexander-Universität Erlangen-Nürnberg
Germany
`{rohan.asthana,vasileios.belagiannis}@fau.de`

## Abstract

Diffusion-based image generative models produce high-fidelity images through iterative denoising but remain vulnerable to memorization, where they unintentionally reproduce exact copies or parts of training images. Recent memorization detection methods are primarily based on the norm of score difference as indicators of memorization. We prove that such norm-based metrics are mainly effective under the assumption of isotropic log-probability distributions, which generally holds at high or medium noise levels. In contrast, analyzing the anisotropic regime reveals that memorized samples exhibit strong angular alignment between the guidance vector and unconditional scores in the low-noise setting. Through these insights, we develop a memorization detection metric by integrating isotropic norm and anisotropic alignment. Our detection metric can be computed directly on pure noise inputs via two conditional and unconditional forward passes, eliminating the need for costly denoising steps. Detection experiments on Stable Diffusion v1.4 and v2 show that our metric outperforms existing denoising-free detection methods while being at least approximately 5x faster than the previous best approach. Finally, we demonstrate the effectiveness of our approach by utilizing a mitigation strategy that adapts memorized prompts based on our developed metric. The code is available at `https://github.com/rohanasthana/memorization-anisotropy`.

## 1 Introduction

Recent advances in diffusion models (Ho et al., 2020; Ho & Salimans, 2021; Rombach et al., 2022), especially score-based models (Song et al., 2021), have positioned them as the dominant class of generative models, synthesizing various types of data, for instance, images (Rombach et al., 2022; Saharia et al., 2022), videos (Ho et al., 2022; Gupta et al., 2024), graphs (Vignac et al., 2023; Liu et al., 2025), and even neural network architectures (Asthana et al., 2024). Score-based diffusion models progressively corrupt data with Gaussian noise and then learn score estimates that approximate the gradients of the perturbed log-density to transform noise back into data. This enables high-quality generation in complex data domains. However, despite the tremendous success of these models, they are susceptible to memorization, where the model generates an exact or near replica of training images. This phenomenon is similar to overfitting in artificial neural networks and has important implications related to data privacy, copyright issues, bias of evaluation benchmarks, and assumptions about generalization. Hence, detection and mitigation of memorization in these high-fidelity generative models has been a growing body of research (Somepalli et al., 2023b; Wen et al., 2024; Chen et al., 2024; Jeon et al., 2025; Jain et al., 2025; Ross et al., 2025). Since the denoising process explicitly interpolates between Gaussian noise and the data manifold, the associated score estimates help to understand the underlying dynamics of diffusion models. This perspective motivates the use of score estimates as a powerful tool to characterize memorization in diffusion models.

Inspired by this perspective, Wen et al. (2024) characterized memorization in text-to-image diffusion models utilizing the norm of the difference of the score between the unconditional and conditional

estimates. This characterization was supported by the observation that memorized samples exhibit stronger text-driven guidance, which guides the generation towards specific memorized images. Later, multiple works adopted this perspective and developed novel methods for detecting and mitigating memorization. For instance, Jeon et al. (2025) improved the metric from Wen et al. (2024) by incorporating the Hessian of the log-probability. Moreover, Jain et al. (2025) utilized Wen's metric to deploy opposite guidance until the denoising trajectory steers away from the memorized sample.

In this work, we demonstrate that such norm-based metrics are effective only when the underlying log-probability is isotropic, which is typically satisfied at high or mid noise levels. This is because the norm of the score function encodes information about the overall curvature of the log-probability. While the overall curvature is a strong signal for memorization in isotropic distributions, it fails in the anisotropic case, as the curvature is different across different directions. This issue is relevant in scenarios where the access to anisotropic diffusion regime is easier/more efficient compared to the isotropic regime. For example, in image level memorization task (Jiang et al., 2025), the aim is to detect memorized images (or parts of them) only through access to generated images. These generated images are far from the high-noise isotropic regime and are at the data manifold. Thus, utilizing a metric that works under the anisotropic regime should be preferable in this case. To overcome this issue, we additionally account for the anisotropy of the log-probability in the low-noise regime and leverage a more informative distribution that better captures memorization than isotropic metrics alone. We explore the anisotropic low-noise regime and reveal that memorization manifests as a stronger alignment between the guidance vector and the unconditional score estimate in the anisotropic low-noise regime. We utilize this phenomenon to develop a simple yet effective memorization detection metric, which is the weighted sum of a) cosine similarity between the guidance vector and the unconditional score estimate in the anisotropic regime, and b) norm of the guidance vector in the isotropic regime. Importantly, our developed metric is denoising-free (meaning that it does not require costly denoising steps) and utilizes only two conditional and unconditional forward passes, hence detecting memorization at a rapid pace. We finally integrate our detection metric into a prompt augmentation scheme to mitigate memorization effectively during inference.

To evaluate the performance and efficiency of our developed metric, we follow the standard evaluation protocol (Wen et al., 2024; Jeon et al., 2025) and conduct experiments on Stable Diffusion v1.4 and v2.0 (Rombach et al., 2022). We show that our method outperforms existing denoising-free metrics while being at least approximately 5x faster than the previous best method. Moreover, our mitigation experiments on MemBench (Hong et al., 2025) showcase that our developed metric makes the generations highly dissimilar to the memorized training sample, while exhibiting strong text-image alignment and high aesthetic quality. Lastly, we conduct three ablation studies in the Appendix Section A.2, comparing alternative formulations of our proposed metric, analyzing the contribution of each component of our metric, and ablating normalization and timestep choices.

In summary, our contributions are as follows:

- We identify that norm-based memorization detection metrics are effective only under the assumption of isotropic log-probability distributions. To rectify this, we consider both the isotropic (high-noise) and anisotropic (low-noise) regimes of the log-probability for memorization detection.

- We show that in the anisotropic regime, memorization manifests as a strong alignment between the guidance vector and the unconditional score estimate. Thus, we develop our denoising-free detection metric by combining the score-based alignment in anisotropy with the norm of the score difference in isotropy.

- Through our experiments, we demonstrate that our metric achieves superior denoising-free detection performance compared to existing metrics while being efficient. We further validate the performance of our developed metric by performing inference-time memorization mitigation through the benchmark MemBench (Hong et al., 2025).

## 2 RELATED WORK

**Memorization in Diffusion Models** Detecting and mitigating memorization in diffusion models has gained substantial interest in recent years (Somepalli et al., 2023b; Wen et al., 2024; Chen et al., 2024; Jeon et al., 2025; Jain et al., 2025; Ross et al., 2025). In text-to-image diffusion models,

this phenomenon was firstly identified in the works by Somepalli et al. (2023a), where they show that factors such as training set size affect the scale of memorization. Concurrently, Carlini et al. (2023) demonstrated the extraction of training data (which were essentially memorized samples) from diffusion models. Subsequent work investigated dataset properties that drive memorization (Kadkhodaie et al., 2023; Pavlova & Wei, 2025), while others proposed guidance-based strategies for mitigation (Jain et al., 2025; Chen et al., 2024). Since the denoising process follows trajectories through the log-probability landscape, another line of work has approached memorization from a geometric perspective. For example, Ross et al. (2025) analyzed the geometry via Local Intrinsic Dimensionality (LID) at the sample-level, and Wen et al. (2024) connected memorization to the norm of score difference, which was later shown by Jeon et al. (2025) to approximate log-probability sharpness in the early denoising phase. Moreover, Chen et al. (2025) utilize this norm-based metric to perform localized memorization detection. While our work also analyses the geometry of the log-probability, it differs fundamentally from previous work because it studies the anisotropic low-noise regime in the context of denoising-free memorization detection. Lastly, concurrent to our work, Brokman et al. (2025) propose a curvature-based criterion that tracks curvature evolution throughout the diffusion trajectory. On the other hand, our approach focuses on first-order angular alignment between conditional and unconditional scores in the anisotropic low-noise regime. Thus, the method from Brokman et al. (2025) can be seen as a higher-order extension of our framework.

**Low-noise regime of Diffusion Models** Several works have analyzed the behavior of diffusion models in the low-noise regime, where the score estimates transition from modeling coarse global structure to capturing fine-grained, data-dependent details (Song et al., 2021; Qian et al., 2024). Recent work shows that the low-noise regime both carries most of the perceptual fidelity and is the regime in which models are most sensitive to dataset overfitting (Qian et al., 2024). Moreover, Pavlova & Wei (2025) systematically study denoising near the data manifold and report that independently trained denoisers can diverge significantly in the low-noise regime. This reveals local inconsistencies that do not appear at high noise levels. Together, these findings suggest that the low-noise regime is a highly informative phase of the denoising process, and thus is a natural setting for studying memorization. Our work builds on this line of research by explicitly characterizing memorization in the low-noise regime and leveraging this for efficient denoising-free memorization detection.

## 3 PRELIMINARIES

**Score-based Diffusion Models** Diffusion models (Sohl-Dickstein et al., 2015; Ho et al., 2020; Song et al., 2021) synthesize data by learning to reverse a gradual noising process. The core idea is to represent data generation as the inversion of a forward stochastic process that maps a complex data distribution, namely $p_0(\mathbf{x})$, to a tractable prior, typically a Gaussian, denoted by $\mathcal{N}(\mathbf{0}, \mathbf{I})$. Let $\mathbf{x}_0 \sim p_0(\mathbf{x})$ be a training data sample. The forward process sequentially corrupts $\mathbf{x}_0$ using the noise model $q$ until the sample reaches a state of pure noise $\mathbf{x}_T \sim \mathcal{N}(\mathbf{0}, \mathbf{I})$, where $T$ is the number of noising timesteps. Formally, a noising step is defined as:

$$q(\mathbf{x}_t|\mathbf{x}_0) = \mathcal{N}(\mathbf{x}_t; \sqrt{\bar{\alpha}_t}\mathbf{x}_0, (1 - \bar{\alpha}_t)\mathbf{I}), \tag{1}$$

where $\mathbf{x}_t$ denotes the noisy data sample at timestep $t \in T$, $\bar{\alpha}_t = \prod_{s=1}^{t}(1 - \beta_s)$, and $\beta_s$ is the diffusion noise schedule at timestep t. In the continuous-time formulation, this process is governed by the forward stochastic differential equation (SDE):

$$d\mathbf{x}_t = f(\mathbf{x}_t, t)dt + g(t)d\mathbf{w}_t, \tag{2}$$

where $f(\cdot)$ is the drift coefficient, $g(\cdot)$ is the diffusion coefficient, and $\mathbf{w}_t$ denotes the standard Brownian motion. The reverse process follows the reverse-time SDE, and is formulated as:

$$d\mathbf{x}_t = \left[ f(\mathbf{x}_t, t) - g^2(t)\nabla_{\mathbf{x}_t} \log p_t(\mathbf{x}_t) \right] dt + g(t)\, d\bar{\mathbf{w}}_t, \tag{3}$$

where $p_t(\mathbf{x}_t)$ is the marginal density of $\mathbf{x}_t$ at time $t$, and $\bar{\mathbf{w}}_t$ is Brownian motion in reverse time. The term $\nabla_{\mathbf{x}_t} \log p_t(\mathbf{x}_t)$ is the gradient of the log-probability density and is defined as a score function

$s(\mathbf{x}_t, t) := \nabla_{\mathbf{x}_t} \log p_t(\mathbf{x}_t)$, which is approximated by a neural network $s_\theta(\mathbf{x}_t, t)$. Upon training, new samples can be generated by simulating the reverse-time SDE using the learned score estimate $s_\theta$.

Additionally, for conditional generation, as in Stable Diffusion (Rombach et al., 2022), one aims to sample from the distribution $p_t(\mathbf{x}_t|c)$, where $c$ is the imposed condition (e.g. a text prompt or a class label). This can be done by estimating the score of the conditional log-probability density $\tilde{s}(\mathbf{x}_t, t, c) := \nabla_{\mathbf{x}_t} \log p_t(\mathbf{x}_t|c)$ through incorporating the condition $c$ in the denoising network $s_\theta$ using classifier-free guidance (Ho & Salimans, 2021). The resulting score estimate is thus defined as:

$$\tilde{s}(\mathbf{x}_t, t, c) = s_\theta(\mathbf{x}_t, t, \varnothing) + w \underbrace{\left[s_\theta(\mathbf{x}_t, t, c) - s_\theta(\mathbf{x}_t, t, \varnothing)\right]}_{\text{conditioning term}}, \tag{4}$$

where $w \geq 1$ controls the strength of the conditioning, $s_\theta(\mathbf{x}_t, t, \varnothing)$ is the unconditional score estimate, $s_\theta(\mathbf{x}_t, t, c)$ is the conditional score estimate and $[s_\theta(\mathbf{x}_t, t, c) - s_\theta(\mathbf{x}_t, t, \varnothing)]$ is the conditioning term. Through Bayes' Theorem, this conditioning term (or guidance vector) essentially corresponds to the gradient of the log-probability $\nabla_{\mathbf{x}_t} \log p_t(c|\mathbf{x}_t)$ (Ho & Salimans, 2021).

**Norm-based Memorization Detection** Memorization in diffusion models occurs when the model reproduces data from the training set either exactly or with only minor variations. In text-to-image diffusion models, this behavior is usually tied to particular text prompts and noise seeds, and can be mitigated by altering the prompt embedding or adjusting the noise initialization utilized in the generation. Most of the recent breakthroughs in prompt-based memorization detection utilize the norm of the score function (Wen et al., 2024; Jeon et al., 2025; Jain et al., 2025). Jeon et al. (2025) showed that such norm-based methods essentially approximate the overall curvature of the underlying log-probability. This is because at medium or high noise levels, the norm of a score function provides an estimate of the trace of the Hessian, which is the sum of all Hessian eigenvalues. This is equal to the overall curvature of the log-probability. One popular metric, utilized by many approaches (Wen et al., 2024; Jain et al., 2025; Jeon et al., 2025), is the norm of the conditioning term, which is the difference between the conditional and unconditional score functions:

$$\|s_\theta^\Delta(\mathbf{x}_t, t, c)\| = \|s_\theta(\mathbf{x}_t, t, c) - s_\theta(\mathbf{x}_t, t)\|. \tag{5}$$

This metric builds on the observation that text prompts that lead to memorized samples (referred to as memorized prompts) exhibit stronger text-driven guidance, quantified by $\|s_\theta^\Delta(\mathbf{x}_t, t, c)\|$. Large values of this norm are therefore indicative of memorization. Jeon et al. (2025) further demonstrate that $\|s_\theta^\Delta(\mathbf{x}_t, t, c)\|$ reflects the extent to which guidance amplifies the curvature of the log-probability. Intuitively, for memorized cases, guidance sharpens the log-probability landscape more aggressively than for non-memorized cases. This results in a sharp peak in the log-probability, which is observed as a signature of memorization.

## 4 METHOD

Consider a conditional diffusion model comprising a neural network $p_\theta$ and trained on the training set $\mathcal{D} = \{x_0^{(i)}\}_{i=1}^N$, where $N$ is the number of training samples. This network is learned to estimate the gradient of the conditional log-probability density $\nabla_{\mathbf{x}_t} \log p_t(\mathbf{x}_t|c)$, using the score function $\tilde{s}(\mathbf{x}_t, t, c)$. Let $c^{mem}$ denote the text prompt, for which denoising through the trained network $p_\theta$ will lead to generating a memorized sample, i.e., an identical sample in the training set associated with the same text prompt. Our objective is to detect $c^{mem}$ without denoising through exploiting the anisotropy of $\log p_t(\mathbf{x}_t|c)$. Furthermore, we aim to mitigate memorization through a prompt augmentation scheme based on our proposed detection metric. To this end, we first motivate our method by explaining the relevance of anisotropy in memorization detection (Sec. 4.1). Next, we demonstrate the emergence of anisotropy in the low-noise regime, along with discussing the failure of previous norm-based detection metrics in anisotropy (Sec. 4.1). To rectify this, we first discuss the signatures of memorization in anisotropy (Sec. 4.2) and propose a novel memorization detection metric using the isotropic curvature and anisotropic angular alignment between the conditional and unconditional score estimates in the low-noise regime (Sec. 4.3). We finally utilize the proposed metric in a memorization mitigation strategy (Sec. 4.3).

## 4.1 ANISOTROPY IN LOG-PROBABILITY

We start by explaining the relevance of anisotropic log-probability in memorization detection. We know that in isotropic distributions, the curvature is the same in every direction. Hence, in the isotropic case, measuring the overall curvature of the log-probability (such as in norm-based methods) is sufficient for memorization detection, as there is no directional variation to exploit. This symmetry means that norm-based methods are inherently unable to extract any additional information beyond how curved or sharp the log-probability is. However, in anisotropic distributions, curvature varies by direction, meaning that some directions in the log-probability are much sharper than others. Thus, we hypothesize that including anisotropic information in memorization detection should yield a better characterization of memorization.

**Anisotropy in Low-Noise Regime**  We now analyze the anisotropy of the conditional log-probability density $\log p_t(\mathbf{x}_t|c)$ in the denoising diffusion process. The curvature of this distribution can be characterized using the Hessian $H(\mathbf{x}_t, c) := \nabla^2_{x_t} \log p_t(\mathbf{x}_t|c)$ (Jeon et al., 2025). We define an isotropic distribution as one that exhibits rotational invariance, implying that $H(\mathbf{x}_t, c)$ is proportional to the identity matrix $\mathbf{I}$, i.e., $H(\mathbf{x}_t, c) = -\lambda\mathbf{I}$, where $\lambda$ is a constant representing the curvature. In this case, all eigenvalues of $H(\mathbf{x}_t, c)$ are equal to $\lambda$, meaning that all directions in $\mathbf{x}_t$ space have identical curvature. Therefore, the variance of the Hessian eigenvalues is almost zero in the isotropic case. Conversely, in the anisotropic case, the curvature $H(\mathbf{x}_t, c)$ varies across directions in the $\mathbf{x}_t$ space, i.e. $H(\mathbf{x}_t, c) = -\mathbf{A}$, where $\mathbf{A}$ is not proportional to the identity, and therefore has unequal eigenvalues. Therefore, the anisotropic case exhibits high variance in the Hessian eigenvalues.

We study in Figure 1 the anisotropy of $\log p_t(\mathbf{x}_t|c)$ by analyzing the variance of the eigenvalues of $H(\mathbf{x}_t, c)$ for each time step in a pre-trained Stable Diffusion model (Rombach et al., 2022) given a random prompt. We observe that the variance of eigenvalues stays minimal during the high-noise regime when $t$ is large, but exhibits larger variance in the low-noise regime, when $t$ is close to 0. This suggests that in the high-noise regime, the log-probability is mostly isotropic, but in low noise, when the density closely estimates the data distribution, the log-probability steers towards anisotropy. The emergence of anisotropy in the low-noise regime motivates our use of score estimates in low noise as indicators of memorization.

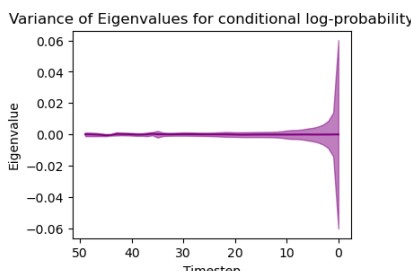

Figure 1: Variance of eigenvalues of the Hessian during denoising.

**Failure of Norm-Based Methods in Anisotropy**  We now demonstrate that norm-based memorization detection methods only work effectively when the underlying log-probability density $\log p_t(\mathbf{x}_t|c)$ is isotropic in nature. Consider the case of a simple conditional isotropic Gaussian log-probability density. In this case, $\log p_t(\mathbf{x}_t|c)$ is defined as:

$$\log p_t(\mathbf{x}_t\,|\,c) = -\frac{1}{2\sigma_t^2}\,\|\mathbf{x}_t - \boldsymbol{\mu}_t(c)\|^2 + C, \tag{6}$$

where $\mu_t(c)$ and $\sigma_t^2$ are the mean and variance of the log-probability at time $t$, and $C$ is a constant. The score function $\tilde{s}(\mathbf{x}_t, t, c)$ estimates the gradient of this distribution, which is:

$$\tilde{s}(\mathbf{x}_t, t, c) = \nabla_{\mathbf{x}_t} \log p_t(\mathbf{x}_t\,|\,c) = -\frac{1}{\sigma_t^2}\,(\mathbf{x}_t - \boldsymbol{\mu}_t(c)). \tag{7}$$

Jeon et al. (2025) show that we can utilize the norm of the score function to characterize curvature, which is defined as:

$$\|\tilde{s}(\mathbf{x}_t, t, c)\| = \frac{1}{\sigma_t^2}\,\|\mathbf{x}_t - \boldsymbol{\mu}_t(c)\|. \tag{8}$$

We observe that the norm $\|\tilde{s}(\mathbf{x}_t, t, c)\|$ in the isotropic case only depends on the variance $\sigma_t^2$ and the distance from the mean $\|\mathbf{x}_t - \boldsymbol{\mu}_t(c)\|$ and not on the direction. Thus, if a memorized sample exhibits a very narrow, peaked density (exhibiting sharp curvature) around some data point, the gradient norm is a sensitive and direct indicator of memorization.

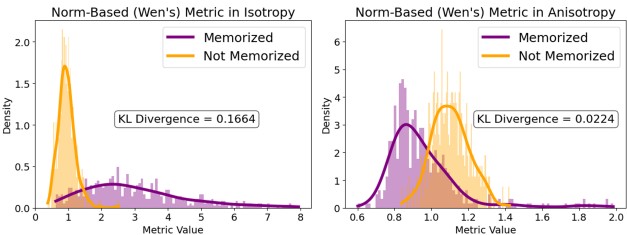

Figure 2: Histograms and Kernel Density Estimation (KDE) curves of Wen's norm-based metric $\|s_\theta^\Delta(\mathbf{x}_t, t, c)\|$ (Wen et al., 2024) in isotropy ($t \approx T$) and anisotropy ($t \approx 0$) under denoising-free inputs ($\mathbf{x}_T$). We observe a larger overlap of KDE curves in anisotropy compared to isotropy, which indicates poor discrimination capabilities between memorized and non-memorized samples.

Now consider the anisotropic log-probability distribution of the form:

$$\log p_t(\mathbf{x}_t \,|\, c) = -\frac{1}{2}(\mathbf{x}_t - \boldsymbol{\mu}_t(c))^T \Sigma_t^{-1}(\mathbf{x}_t - \boldsymbol{\mu}_t(c)) + C, \tag{9}$$

where $\Sigma_t$ is the covariance matrix with non-identical eigenvalues. Calculating the score function by taking the gradient of the log-probability, we get:

$$\tilde{s}(\mathbf{x}_t, t, c) = \nabla_{\mathbf{x}_t} \log p_t(\mathbf{x}_t \,|\, c) = -\Sigma_t^{-1}(\mathbf{x}_t - \boldsymbol{\mu}_t(c)). \tag{10}$$

Finally, taking the norm of the score function, we get:

$$\|\tilde{s}(\mathbf{x}_t, t, c)\| = \sqrt{(\mathbf{x}_t - \boldsymbol{\mu}_t(c))^T \Sigma_t^{-2}(\mathbf{x}_t - \boldsymbol{\mu}_t(c))}. \tag{11}$$

Since the covariance matrix has non-identical eigenvalues, we observe that $\|\tilde{s}(\mathbf{x}_t, t, c)\|$ depends on both the direction and the distance from the mean. Thus, in anisotropy, a high norm in one direction can be compensated by a low norm in another direction. Hence, the overall norm may not spike even if memorization is present, which might lead to false negatives. Therefore, norm-based methods are only effective under the assumption of isotropy in the underlying log-probability distribution. We experimentally validate this finding through plotting the histograms and Kernel Density Estimation (KDE) curves of the norm-based metric by Wen et al. (2024), i.e., $\|s_\theta^\Delta(\mathbf{x}_t, t, c)\|$ for memorized and non-memorized cases in both isotropy and anisotropy. We additionally compare the Kullback–Leibler (KL) divergence of memorized and non-memorized distributions in each case. We observe from Figure 2 that in isotropy, KDE curves have less overlap with a high KL Divergence (=0.166), indicating better discriminating capabilities of $\|s_\theta^\Delta(\mathbf{x}_t, t, c)\|$. In contrast, this overlap is higher in anisotropy with a low KL Divergence (=0.022), thus depicting the failure of norm-based metrics in the low-noise anisotropic case.

## 4.2 MEMORIZATION THROUGH ANGULAR ALIGNMENT

We now discuss memorization signatures in the low-noise anisotropic regime of the denoising process. Consider the denoising process of a diffusion model using score estimates (Eq. 4). By score decomposition, the denoising score under guidance can be written as

$$s_\theta(\mathbf{x}_t, t, c; w) = \nabla_{\mathbf{x}_t} \log p_t(\mathbf{x}_t) + w \nabla_{\mathbf{x}_t} \log p_t(c|\mathbf{x}_t). \tag{12}$$

Wen et al. (2024) show that in memorized cases, the conditioning term $\nabla_{\mathbf{x}_t} \log p_t(c|\mathbf{x}_t)$ has significantly larger norm than in non-memorized cases. Under guidance, this term is amplified by the guidance weight $w$, meaning that for memorized samples, the denoising trajectory is dominated by the conditional score rather than the unconditional score. Thus, the reverse process is biased toward the memorized mode instead of the broader data distribution. This behaviour is also evident in the work from Jain et al. (2025) where they show that applying guidance after a certain timestep prevents memorization. This is because if the guidance is eliminated during the early stages of denoising of a memorized case, the unconditional gradient $\nabla_{\mathbf{x}_t} \log p_t(\mathbf{x}_t)$ steers the trajectory away from the memorized mode until the nearest mode for $\nabla_{\mathbf{x}_t} \log p_t(c|\mathbf{x}_t)$ is not a memorized one.

This means that in the later stages of denoising (low-noise regime) of memorized cases, the nearest mode for both $\log p_t(\mathbf{x}_t)$ and $\log p_t(c|\mathbf{x}_t)$ should correspond to the memorized mode, hence the conditioning should merely reinforce the direction of the unconditional gradient without introducing new directions. To this end, consider the following theorem:

**Theorem 1** *Consider the anisotropic low-noise regime of diffusion and let $\Sigma_t, \Sigma_t^c$ be symmetric positive definite and $v_t := \mathbf{x}_t - \mu$ and $\delta := \mu_c - \mu$ denote the sample displacement from the unconditional mode and the relative displacement between the guidance mode and unconditional mode. Then,*

$$s_\theta(\mathbf{x}_t, t) = \nabla_{\mathbf{x_t}} \log p_t(\mathbf{x}_t) := -\Sigma_t^{-1} v_t; \quad s_\theta^\Delta(\mathbf{x}_t, t, c) = \nabla_{\mathbf{x_t}} \log p_t(c|\mathbf{x}_t) := (\Sigma_t^{-1} - \Sigma_t^{c^{-1}}) v_t + \Sigma_t^{c^{-1}} \delta.$$

*Let $\alpha > 0$ and constants $\varepsilon, \tau \geq 0$. Assume*

$$\|(\Sigma_t^{-1} - \Sigma_t^{c^{-1}}) v_t - \alpha s_\theta(\mathbf{x}_t, t)\| \leq \varepsilon \alpha \|s_\theta(\mathbf{x}_t, t)\|, \qquad \|\Sigma_t^{c^{-1}} \delta\| \leq \tau \alpha \|s_\theta(\mathbf{x}_t, t)\|,$$

*and set $r := \varepsilon + \tau < 1$. Then the cosine similarity satisfies*

$$\cos(s_\theta(\mathbf{x}_t, t), s_\theta^\Delta(\mathbf{x}_t, t, c)) \geq \frac{1 - r}{1 + r}. \tag{13}$$

The proof of this theorem is provided in the Appendix Section A.1. This theorem provides a lower-bound for the cosine similarity between $\nabla_{\mathbf{x}_t} \log p_t(\mathbf{x}_t)$ and $\nabla_{\mathbf{x}_t} \log p_t(c|\mathbf{x}_t)$ using the relative displacement ($\delta$) between the guidance mode and the unconditional mode. Specifically, if both the unconditional and guidance modes nearly coincide ($\delta \to 0$), the only deviation between $s_\theta^\Delta(\mathbf{x}_t, t, c)$ and scaled $s_\theta(\mathbf{x}_t, t)$ arises from differences in covariance, which are controlled by the approximation error $\varepsilon$. The resulting error term $r = \varepsilon + \tau$ simplifies to $r = \varepsilon$, and the cosine similarity lower bound in the theorem becomes $\frac{1-\varepsilon}{1+\varepsilon}$. Thus, if $\varepsilon$ is small, the alignment between $\nabla_{\mathbf{x}_t} \log p_t(\mathbf{x}_t)$ and $\nabla_{\mathbf{x}_t} \log p_t(c|\mathbf{x}_t)$ becomes high.

**Remark 1** *Memorized cases exhibit small $\delta$ and thus higher angular alignment between $\nabla_{\mathbf{x}_t} \log p_t(\mathbf{x}_t)$ and $\nabla_{\mathbf{x}_t} \log p_t(c|\mathbf{x}_t)$ in the anisotropic low-noise regime compared to non-memorized cases.*

We empirically verify this by examining the angular alignment between the conditioning term ($\nabla_{\mathbf{x}_t} \log p_t(c|\mathbf{x}_t)$) and unconditional log-probability ($\nabla_{\mathbf{x}_t} \log p_t(\mathbf{x}_t)$) in the anisotropic low-noise regime ($t \approx 0$) for both memorized and non-memorized samples. We utilize Stable Diffusion v1.4 for this analysis and plot the directions of respective gradients (Figure 3a) and the cosine similarity between these gradients in the form of a heatmap (Figure 3b). We observe that memorized samples exhibit significantly higher alignment, and thus, intuitively, higher cosine similarity between the conditioning term and the unconditional score estimates. This means that for the memorized cases in anisotropy, conditioning does not introduce new directions to the unconditional gradient. In contrast, non-memorized cases generally exhibit misaligned directions, implying weak or random angular correlation between $\nabla_{\mathbf{x}_t} \log p_t(\mathbf{x}_t)$ and $\nabla_{\mathbf{x}_t} \log p_t(c|\mathbf{x}_t)$. This behavior aligns with our theoretical prediction: in anisotropic neighborhoods surrounding memorized data, the conditional log-probability mode coincides with the mode of unconditional log-probability, producing high angular alignment.

### 4.3 DETECTION METRIC AND MITIGATION

**Detection** We know that memorization in diffusion models manifests differently across the anisotropic and isotropic regimes of the denoising trajectory. Combining these regimes should allow us to exploit a more informative log-probability, which in turn should improve the characterization of memorization. Hence, we formulate our denoising-free memorization detection metric incorporating both regimes of the denoising process. For the case of the isotropic high-noise regime, we utilize the norm of the score difference $\|s_\theta^\Delta(\mathbf{x}_t, t, c)\|$ (Wen et al., 2024), and for the case of anisotropic low-noise regime, we utilize the angular alignment (calculated using cosine similarity) between the conditioning term $s_\theta^\Delta(\mathbf{x}_t, t, c)$ and the unconditional score estimate $s_\theta(\mathbf{x}_t, t)$. Finally, we combine both metrics through a weighted sum:

$$\mathcal{M}(\mathbf{x}_T, c) = \gamma_1 \underbrace{\left\{ \frac{\langle s_\theta^\Delta(\mathbf{x}_T, t \approx 0, c), \, s_\theta(\mathbf{x}_T, t \approx 0) \rangle}{\|s_\theta^\Delta(\mathbf{x}_T, t \approx 0, c)\| \, \|s_\theta(\mathbf{x}_T, t \approx 0)\|} \right\}}_{\text{cosine similarity in anisotropy}} + \gamma_2 \underbrace{\|s_\theta^\Delta(\mathbf{x}_T, t \approx T, c)\|}_{\text{norm of score difference in isotropy}}, \tag{14}$$

where $\gamma_1$ and $\gamma_2$ are parameters controlling the weight of each term, and $\langle \cdot, \cdot \rangle$ denotes the dot product. Importantly, we calculate our metric at artificially set timesteps $t = 0$ and $t = T$ using the same initial Gaussian noise sample $\mathbf{x}_T \sim \mathcal{N}(\mathbf{0}, \mathbf{I})$, i.e., we do not run a reverse denoising trajectory and simply query the model at different noise levels.

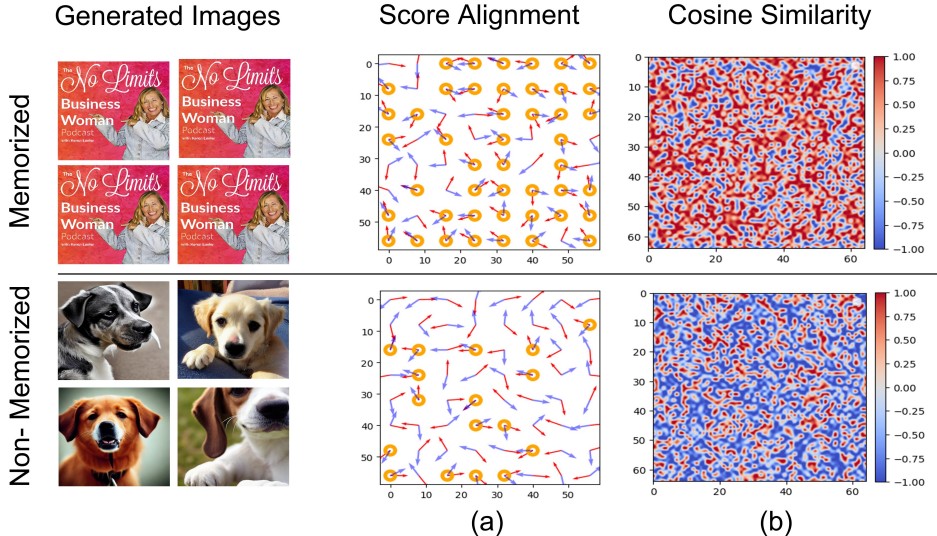

Figure 3: Comparison of angular alignment between $\nabla_{\mathbf{x}_t} \log p_t(\mathbf{x}_t)$ and $\nabla_{\mathbf{x}_t} \log p_t(c|\mathbf{x}_t)$, along with the heatmap of cosine similarity between them, for memorized and non-memorized cases. **(a)** We observe a larger number of highly-aligned vectors for the memorized case compared to the non-memorized case, indicated through orange rings. **(b)** We observe generally a higher cosine similarity (indicated as red regions) in the memorized case compared to the non-memorized case.

**Inference-time Mitigation**    Similar to Wen et al. (2024); Ross et al. (2025), we mitigate memorization during inference through the prompt augmentation technique proposed by Wen et al. (2024). Specifically, we optimize text prompt embeddings at initialization through gradient descent using our detection metric as a loss. Thus, our loss is defined as:

$$\mathcal{L}(\mathbf{x}_T, c) = \mathcal{M}(\mathbf{x}_T, c) \tag{15}$$

After optimization, we obtain a prompt embedding $c^\star$, which we utilize for generating non-memorized data through the denoising process.

## 5 EXPERIMENTS

Following the related work (Jeon et al., 2025; Ross et al., 2025; Jain et al., 2025), we evaluate our method under two standard tasks, namely memorization detection and inference-time mitigation. Our detection evaluation includes experiments on Stable Diffusion (SD) v1.4 and v2.0 (Rombach et al., 2022), whereas our mitigation evaluation is performed through the recently developed memorization benchmark MemBench (Hong et al., 2025). Additionally, we conduct an ablation study in Appendix Section A.2, where we compare different formulations of our metric.

### 5.1 MEMORIZATION DETECTION

**Experimental Setup**    Following Jeon et al. (2025), we perform our detection experiments by utilizing 500 memorized text prompts for SD v1.4 and 219 memorized prompts for SD v2 provided by Webster (2023). We additionally include 500 non-memorized prompts from Lexica (Shen et al., 2024), GPT-4 (Achiam et al., 2023), COCO (Lin et al., 2014), and Tuxemon (Tsaban & Paul, 2024), identical to Jeon et al. (2025). Through these prompts, we calculate our detection metric (Eq. 14) using two forward passes of the model (for $t = 1$ and $t = T$) for 3 different runs, each with a different seed and report the mean and Standard Deviation (StD) of our results. We assess our detection metric through two standard measures, namely the Area Under the Receiver Operating Characteristic Curve (AUC) and the True Positive at 1% False Positive Rate (TPR@1%FPR). Additionally, we also report the time taken to calculate the metrics for 10 prompts in seconds. We consider two cases for our experimentation, each with a different number of generations ($n$). We compare our method with several denoising-free detection baselines, including the metric proposed by Ren et al. (2024)

Table 1: Comparison of denoising-free memorization detection methods on SD v1.4 and SD v2.0. We calculate our metric for 3 runs, each with a different seed, and report the mean $\pm$ standard deviation (StD). Here, $n$ represents the number of generations and Time (sec.) represents the time taken to calculate the metric for 10 prompts in seconds. All results except Time are taken from Jeon et al. (2025). The best numbers are indicated as **bold** and the second best numbers are indicated as underline.

| Method | SD v1.4 | | | SD v2.0 | | |
|---|---|---|---|---|---|---|
| | AUC $\uparrow$ | TPR@1%FPR $\uparrow$ | Time (sec.) $\downarrow$ | AUC $\uparrow$ | TPR@1%FPR $\uparrow$ | Time (sec.) $\downarrow$ |
| $n = 1$ | | | | | | |
| Ren et al. (2024) | 0.846 | 0.116 | 0.05 | 0.848 | 0 | 0.07 |
| Wen et al. (2024) | 0.976 | 0.896 | 0.40 | 0.948 | 0.739 | 0.80 |
| Jeon et al. (2025) | 0.987 | 0.908 | 5.40 | **0.959** | 0.740 | 14.60 |
| $\mathcal{M}(\mathbf{x_T}, \mathbf{c})$ (ours) | **0.994 $\pm$0.001** | **0.935$\pm$0.002** | 1.10 | 0.953$\pm$0.016 | **0.791$\pm$0.015** | 2.20 |
| $n = 4$ | | | | | | |
| Ren et al. (2024) | 0.839 | 0.130 | 0.05 | 0.853 | 0 | 0.07 |
| Wen et al. (2024) | 0.992 | 0.944 | 1.20 | 0.980 | 0.876 | 2.70 |
| Jeon et al. (2025) | 0.998 | 0.982 | 19.40 | **0.991** | **0.895** | 56.40 |
| $\mathcal{M}(\mathbf{x_T}, \mathbf{c})$ (ours) | **0.999$\pm$0.001** | **0.984$\pm$0.002** | 3.40 | 0.981$\pm$0.003 | 0.890 $\pm$ 0.009 | 7.30 |

utilizing cross-attention in text-conditioning, the norm-based metric proposed by Wen et al. (2024), and lastly the sharpness-based metric by Jeon et al. (2025). We perform additional experiments on Realistic Vision (CivitAI, 2023) in Appendix Section A.6. Additional experimental details are provided in Appendix Section A.3.

**Discussion of Results** Table 1 presents a comparison of denoising-free memorization detection methods on SD v1.4 and SD v2.0 across different numbers of generations $n$. On SD v1.4, our method consistently surpasses previous methods across all metrics for both $n = 1$ and $n = 4$. Specifically, it achieves state-of-the-art AUC scores of 0.994 ($n = 1$) and 0.999 ($n = 4$), along with the highest TPR@1%FPR values of 0.935 ($n = 1$) and 0.984 ($n = 4$). Furthermore, it provides a speedup of 4.91x ($n = 1$) and 5.71x ($n = 4$) over the next best method (Jeon et al., 2025). This is because, unlike the method from Jeon et al. (2025), ours does not require costly calculations of the Hessian of the log-probability. On SD v2.0, Jeon et al. (2025) attains slightly higher AUC scores of 0.959 ($n = 1$) and 0.991 ($n = 4$) compared to our method's 0.953 ($n = 1$) and 0.981 ($n = 4$). However, our approach improves TPR@1%FPR by 5.1% in $n = 1$ case. In addition, it offers a runtime speedup of 6.63x ($n = 1$) and 7.73x ($n = 4$) over the previous best method. This showcases that our method possesses stronger detection capabilities under strict false-positive constraints and is very efficient. Moreover, we observe that the StD values of TPR@1%FPR are generally higher than AUC, indicating that TPR@1%FPR is more sensitive to random seed. Lastly, although the approach from Wen et al. (2024) is on average $\approx 0.2$ seconds faster per prompt than our approach, our method shows improvements of TPR@1%FPR by up to 5.2% (average improvement of 3.6%) compared to Wen et al. (2024), indicating better discriminating capabilities in edge cases. This is because our method combines independent predictors of memorization and thus, is more robust to these edge cases. We argue that this increment is critical, especially in scenarios where minimizing false negatives is more important than compromising the detection speed by $\approx 0.2$ seconds. Overall, these results demonstrate that incorporating anisotropy for memorization detection significantly improves the detection performance and offers a more reliable approach under strict false-positive constraints, while also exhibiting a rapid pace.

## 5.2 MEMORIZATION MITIGATION

**Experimental Setup** We conduct our quantitative and qualitative evaluation on inference-time mitigation through MemBench (Hong et al., 2025). It utilizes a standardized prompt augmentation approach (as described in Section 4.3) for a fair comparison of memorization mitigation strategies. We assess the mitigation strategy on SD v1.4 and SD v2.0 through our proposed metric by computing several metrics, namely SSCD Similarity Score (Pizzi et al., 2022) to estimate the similarity between the generated sample and the memorized training sample, CLIP Score (Radford et al., 2021) to assess the text-image alignment, and Aesthetic Score (Schuhmann et al., 2022) to evaluate the quality of the generated image. For SD v1.0, we consider 3000 memorized prompts provided by Hong et al. (2025) and for SD v2.0, we utilize 219 prompts provided by Webster (2023). We conduct our mitigation

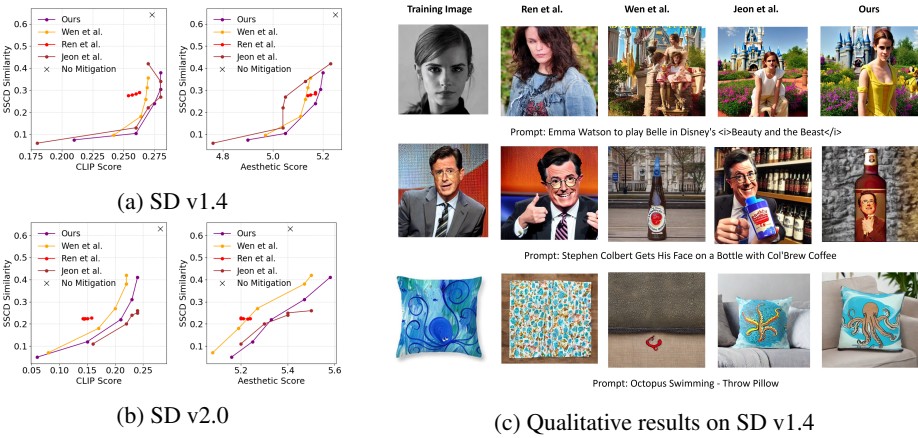

Figure 4: (**a,b**) Quantitative comparison of inference-time mitigation methods on SD v1.4 and SD v2.0. The evaluation is done across five distinct hyperparameter configurations. Lower values of SSCD Similarity and higher values of CLIP Score and Aesthetic Score are desirable. (**c**) Qualitative comparison of inference-time mitigation strategies on SD v1.4.

experiment a total of five times, each with a different hyperparameter configuration. We compare the mitigation strategy through our metric with no mitigation, along with mitigation methods from Wen et al. (2024), Ren et al. (2024), and Jeon et al. (2025). Additional details can be found in Appendix Section A.3.

**Discussion of Results**   The results of our mitigation experiments are shown in Figure 4. Our quantitative results show a general trend demonstrated by all metrics that higher SSCD Similarity corresponds to a higher CLIP and Aesthetic Score, thus exhibiting a tradeoff between these metrics. Ideally, lower SSCD similarity with higher CLIP and Aesthetic score is favourable. For both SD v1.4 and SD v2.0, our proposed metric achieves a good trade-off between CLIP score and SSCD similarity, as well as between Aesthetic score and SSCD similarity. In particular, our method attains a significantly lower similarity score than Wen et al. (2024) and Ren et al. (2024) while maintaining the same text-image alignment (CLIP score) as well as image quality (Aesthetic Score). Morover, Jeon et al. (2025) attains similar CLIP and worse Aesthetic score in SD v1.4, while attaining slightly higher CLIP and Aesthetic score in SD v2.0, but is also at least 7x slower (as shown in Table 1). In general, our results prove the effectiveness of our method in producing non-memorized high-quality images that are well-aligned with the text prompt. Additionally, our qualitative results demonstrate that our approach mitigates memorization more effectively than prior methods, generating high-quality images that are aligned with the text prompt and clearly distinct from the training image. We provide an exhaustive set of qualitative comparisons of our mitigation approach with the baselines in the Appendix Section A.7.

## 6   CONCLUSION

In this work, we demonstrated that norm-based memorization detection metrics are only reliable under the assumption of isotropic log-probability distributions, which generally holds at high or medium noise levels. This is because the norm-based metrics encode the overall sharpness of the log-probability, while in anisotropic settings, the sharpness varies across directions, making the norm-based metrics ineffective. To address this, we proposed a denoising-free detection metric that leverages both the isotropic and anisotropic nature of the log-probability. Our metric is composed of the weighted sum of a) the norm of the guidance vector in isotropy and b) the cosine similarity between the guidance vector and unconditional score estimates in anisotropy. Results from detection experiments on Stable Diffusion v1.4 and v2 show that our method outperforms prior denoising-free approaches while remaining at least approximately 5x faster than the previous best approach. Lastly, inference-time mitigation experiments on Stable Diffusion v1.4 and v2 reveal that our approach achieves a more favorable CLIP/Aesthetic and SSCD similarity trade-off, generating high-quality, non-memorized images that remain well aligned with the input text prompt.

ACKNOWLEDGMENT

We are sincerely thankful to the German National Science Foundation (DFG) for supporting and funding this work under the project *Always-on Deep Neural Networks* (grant number BE 7212/7-1 — OR245/19-1). We additionally acknowledge the Erlangen National High Performance Computing Center (NHR@FAU) of the Friedrich-Alexander-Universität Erlangen-Nürnberg (FAU) for providing computing resources.

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

# A APPENDIX

## A.1 PROOF FOR THEOREM 1

By definition,

$$s_\theta^\Delta(\mathbf{x}_t, t, c) = (\Sigma_t^{-1} - \Sigma_t^{c^{-1}})v_t + \Sigma_t^{c^{-1}}\delta. \tag{16}$$

By assumption, there exists $\alpha > 0$ such that

$$(\Sigma_t^{-1} - \Sigma_t^{c^{-1}})v_t = \alpha s_\theta(\mathbf{x}_t, t) + \Delta_1, \qquad \|\Delta_1\| \le \varepsilon\alpha\|s_\theta(\mathbf{x}_t, t)\|, \tag{17}$$

and

$$\Sigma_t^{c^{-1}}\delta = \Delta_2, \qquad \|\Delta_2\| \le \tau\alpha\|s_\theta(\mathbf{x}_t, t)\|. \tag{18}$$

Thus,

$$s_c = \alpha s_\theta(\mathbf{x}_t, t) + \Delta_1 + \Delta_2 = \alpha s_\theta(\mathbf{x}_t, t) + \Delta, \tag{19}$$

where $\Delta = \Delta_1 + \Delta_2$ satisfies

$$\|\Delta\| \le (\varepsilon + \tau)\alpha\|s_\theta(\mathbf{x}_t, t)\| = r\alpha\|s_\theta(\mathbf{x}_t, t)\|. \tag{20}$$

The cosine similarity is

$$\cos(s_\theta(\mathbf{x}_t, t), s_\theta^\Delta(\mathbf{x}_t, t, c)) = \frac{\langle s_\theta(\mathbf{x}_t, t), \alpha s_\theta(\mathbf{x}_t, t) + \Delta \rangle}{\|s_\theta(\mathbf{x}_t, t)\| \|\alpha s_\theta(\mathbf{x}_t, t) + \Delta\|}. \tag{21}$$

For the numerator,

$$\begin{aligned}
\langle s_\theta(\mathbf{x}_t, t), \alpha s_\theta(\mathbf{x}_t, t) + \Delta \rangle &= \alpha\|s_\theta(\mathbf{x}_t, t)\|^2 + \langle s_\theta(\mathbf{x}_t, t), \Delta \rangle \\
&\ge \alpha\|s_\theta(\mathbf{x}_t, t)\|^2 - \|s_\theta(\mathbf{x}_t, t)\|\|\Delta\| \\
&\ge (1 - r)\alpha\|s_\theta(\mathbf{x}_t, t)\|^2.
\end{aligned} \tag{22}$$

For the denominator,

$$\|\alpha s_\theta(\mathbf{x}_t, t) + \Delta\| \le \alpha\|s_\theta(\mathbf{x}_t, t)\| + \|\Delta\| \le (1 + r)\alpha\|s_\theta(\mathbf{x}_t, t)\|. \tag{23}$$

Therefore,

$$\cos(s_\theta(\mathbf{x}_t, t), s_\theta^\Delta(\mathbf{x}_t, t, c)) \ge \frac{(1 - r)\alpha\|s_\theta(\mathbf{x}_t, t)\|^2}{\|s_\theta(\mathbf{x}_t, t)\|(1 + r)\alpha\|s_\theta(\mathbf{x}_t, t)\|} = \frac{1 - r}{1 + r}. \tag{24}$$

This completes the proof.

## A.2 ABLATION STUDIES

### A.2.1 COMPARING DIFFERENT FORMULATIONS

We now compare different formulations of our detection metric. For this, we perform our detection experiments on SD v1.4 and SD v2.0 (discussed in Section 5.1) on various formulations. Specifically, we consider the cosine similarity between a) $\nabla \log p_t(\mathbf{x}_t)$ and $\nabla \log p_t(c|\mathbf{x}_t)$ (original formulation); b) $\nabla \log p_t(\mathbf{x}_t)$ and $\nabla \log p_t(\mathbf{x}_t|c)$; and lastly c) $\nabla \log p_t(\mathbf{x}_t|c)$ and $\nabla \log p_t(c|\mathbf{x}_t)$. Identical to the Section 5.1, we report the Area Under the Receiver Operating Characteristic Curve (AUC), along with the True Positive at 1% False Positive Rate (TPR@1%FPR) for two cases, namely for number of generations $n = 1$ and $n = 4$. The results are reported in Table 2.

We observe that the formulations $cos(\nabla \log p_t(\mathbf{x}_t|c), \nabla \log p_t(c|\mathbf{x}_t))$ and $cos(\nabla \log p_t(\mathbf{x}_t), \nabla \log p_t(c|\mathbf{x}_t))$ (original formulation) achieve the best overall performance, however the difference between all formulations is only marginal. Importantly, the best two formulations achieve exactly the same performance. This is because as we can observe from Eq. 12, the conditional score $s_\theta(\mathbf{x}_t, t, c; w) = \nabla \log p_t(\mathbf{x}_t|c)$ differs from the unconditional score $\nabla \log p_t(\mathbf{x}_t)$ only by the term $\nabla \log p_t(c|\mathbf{x}_t)$, which has high alignment with $\nabla \log p_t(c|\mathbf{x}_t)$. This causes both formulations to end up capturing essentially the same directional information, leading to identical results.

Table 2: Comparison of different formulations of our proposed metric on SD v1.4 and SD v2.0. Here, $n$ represents the number of generations.

| Method | SD v1.4 | | SD v2.0 | |
|---|---|---|---|---|
| | AUC ↑ | TPR@1%FPR ↑ | AUC ↑ | TPR@1%FPR ↑ |
| $n = 1$ | | | | |
| $cos(\nabla \log p_t(\mathbf{x}_t), \nabla \log p_t(\mathbf{x}_t|c))$ | 0.980 | 0.908 | 0.952 | 0.753 |
| $cos(\nabla \log p_t(\mathbf{x}_t|c), \nabla \log p_t(c|\mathbf{x}_t))$ | 0.992 | 0.934 | 0.952 | 0.749 |
| $cos(\nabla \log p_t(\mathbf{x}_t), \nabla \log p_t(c|\mathbf{x}_t))$ (original) | 0.992 | 0.934 | 0.952 | 0.749 |
| $n = 4$ | | | | |
| $cos(\nabla \log p_t(\mathbf{x}_t), \nabla \log p_t(\mathbf{x}_t|c))$ | 0.993 | 0.940 | 0.980 | 0.904 |
| $cos(\nabla \log p_t(\mathbf{x}_t|c), \nabla \log p_t(c|\mathbf{x}_t))$ | 0.999 | 0.984 | 0.981 | 0.900 |
| $cos(\nabla \log p_t(\mathbf{x}_t), \nabla \log p_t(c|\mathbf{x}_t))$ (original) | 0.999 | 0.984 | 0.981 | 0.900 |

### A.2.2 CONTRIBUTION OF EACH COMPONENT

We perform an ablation study to assess the individual contributions of the two components in our metric, i.e., the norm of the score difference in isotropy and the cosine similarity in anisotropy. Specifically, we evaluate two modified configurations of the metric, one where we set $\gamma_1 = 0$, and another where we set $\gamma_2 = 0$. For each configuration, we run detection experiments following the same protocol described in Section 5.1. We report the AUC and TPR@1%FPR for ($n = 1$) and ($n = 4$), summarized in Table 3. We observe that the norm of the score difference generally performs better than cosine similarity. However, the combination of the two terms exceeds the performance of both terms individually. Specifically, we find that the cosine similarity performs worse in SD v2.0 compared to SD v1.4. This is because the memorized prompt set for SD v2.0 consists mostly of local memorization cases, where only some parts of training set are memorized. In these cases, the mode displacement $\delta$ between $\log p_t(\mathbf{x}_t)$ and $\log p_t(c|\mathbf{x}_t)$ is large because the other non-memorized features increase the mode distance. Hence, the cosine similarity becomes lower, and the alignment is no longer a reliable metric. Therefore, the combination of these two metrics is necessary for robust local memorization detection and mitigation.

Table 3: Comparison of individual components of our proposed metric on SD v1.4 and SD v2.0. Here, $n$ represents the number of generations.

| Method | SD v1.4 | | SD v2.0 | |
|---|---|---|---|---|
| | AUC ↑ | TPR@1%FPR ↑ | AUC ↑ | TPR@1%FPR ↑ |
| $n = 1$ | | | | |
| Norm of the score difference in isotropy | 0.976 | 0.896 | 0.948 | 0.739 |
| Cosine similarity in anisotropy | 0.923 | 0.424 | 0.779 | 0.416 |
| **Combined (ours)** | 0.992 | 0.934 | 0.952 | 0.749 |
| $n = 4$ | | | | |
| Norm of the score difference in isotropy | 0.992 | 0.944 | 0.980 | 0.876 |
| Cosine similarity in anisotropy | 0.939 | 0.440 | 0.785 | 0.401 |
| **Combined (ours)** | 0.999 | 0.984 | 0.981 | 0.900 |

### A.2.3 ABLATING NORMALIZATION AND TIMESTEP

We now conduct an ablation where we analyze the robustness of our approach when using different normalization methods and different timesteps $t$ for computing cosine similarity. Specifically, we consider L1, L2 and spatial L2 normalization, along with timesteps $t = 1, 2, 3$. Here $t$ represents the DDIM timestep. We utilize SD v1.4 and SD v2.0 for these experiments for $n = 1$ case. The results can be observed in Table 4. The results demonstrate that our approach is robust across all the cases in both SD v1.4 and SD v2.0.

## A.3 EXPERIMENTAL DETAILS

### A.3.1 FIGURE 2 EXPERIMENT

To empirically demonstrate the failure of norm-based methods in anisotropy, we conduct the Figure 2 experiment comparing Histograms, KDE curves, and KL Divergence with respect to two cases,

Table 4: Our metric on different normalizations and timesteps on SD v1.4 and SD v2.0 for $n = 1$.

| Method | SD v1.4 | | SD v2.0 | |
|---|---|---|---|---|
| | AUC ↑ | TPR@1%FPR ↑ | AUC ↑ | TPR@1%FPR ↑ |
| Normalization | | | | |
| L1 | 0.997 | 0.950 | 0.937 | 0.804 |
| L2 | 0.997 | 0.950 | 0.937 | 0.804 |
| Spatial L2 | 0.995 | 0.934 | 0.932 | 0.806 |
| Timesteps | | | | |
| t=1 | 0.993 | 0.934 | 0.953 | 0.800 |
| t=2 | 0.987 | 0.862 | 0.933 | 0.794 |
| t=3 | 0.985 | 0.860 | 0.933 | 0.805 |

isotropy and anisotropy. For both isotropy (t ≈ T) and anisotropy (t ≈ 0), we consider a denoising-free scenario, i.e., we utilize pure noise $\mathbf{x_T}$ as inputs to calculate Wen's metric. The experiment is conducted on Stable Diffusion v1.4, and we utilize the same 500 memorized prompts and 500 non-memorized prompts as in the detection experiments. In anisotropy, we clip the values of the metric larger than 2.0 to improve the visibility of the visualization. The KDE and KL Divergence are calculated using the SciPy library, and plots are generated using Matplotlib.

### A.3.2 DETECTION

We run our detection experiments with Python 3.11.5 on a single NVIDIA RTX A6000 GPU with 48GB VRAM. Moreover, we utilize DDIM Sampler (Song et al., 2020) for denoising. For identifying the optimal values of $\gamma_1$ and $\gamma_2$ for SD v1.4 and SD v2.0, we fit a simple Logistic Regressor once on a small set (of size 20) of memorized prompts. Then, we use the found optimal values for all of our detection and mitigation experiments. We utilize the diffusers library with *'CompVis/stable-diffusion-v1-4'* for loading SD v1.4 and *'stabilityai/stable-diffusion-2'* for loading SD v2.0 from HuggingFace. Our evaluation protocol for detection is identical to the one provided by Jeon et al. (2025). For reporting the time taken by metrics, we consider the time between the start of the first forward pass of the model and when the metric has been calculated. In Table 5, we provide the hyperparameter values for all of our detection experiments.

Table 5: Hyperparameter values for detection experiments.

| Hyperparameter | SD v1.4 | SD v2.0 |
|---|---|---|
| $\gamma_1$ | 2.0 | 0.1 |
| $\gamma_2$ | 1.0 | 1.0 |
| Random Seed | 42 | 42 |
| Guidance Scale ($w$) | 7.5 | 7.5 |
| Inference Steps ($T$) | 50 | 50 |

### A.3.3 MITIGATION

Our mitigation experiments are conducted using Python 3.11.5 on a single NVIDIA RTX A6000 GPU. We utilize the memorization mitigation benchmark MemBench (Hong et al., 2025), which provides a standard evaluation for all the methods for SD v1.4 and SD v2.0. We run the mitigation experiments on our approach with 5 distinct hyperparameter configurations. The configurations are provided in Table 6.

### A.4 EXPLANATION ON TIMESTEP MISMATCH

A natural concern is why our metric remains informative when we deliberately evaluate the model at $t \approx 0$. (low-noise regime) while inputting a high-noise latent $\mathbf{x}_T$. In principle, this creates a mismatch, i.e., the model is conditioned on an early timestep, but the provided input $\mathbf{x}_T$ is far from the data manifold. We argue that one plausible explanation for why probing a model at $t = 0$ while inputting high noise works so well in memorization detection is because memorization is encoded in the learned log-probability of the trained model and is mostly independent of the input sample

Table 6: Five distinct hyperparameter configurations for our mitigation experiments on SD v1.4 and SD v2.0.

| Config | $n$ | Learning Rate | $T$ | Iterations | Optimal Loss | Guidance Scale ($w$) |
|--------|-----|---------------|-----|------------|--------------|----------------------|
| SD v1.4 | | | | | | |
| 1 | 1 | 0.05 | 50 | 50 | -1.0 | 7.5 |
| 2 | 1 | 0.05 | 50 | 50 | -0.4 | 7.5 |
| 3 | 1 | 0.05 | 50 | 2 | -0.2 | 7.5 |
| 4 | 1 | 0.05 | 50 | 1 | -0.2 | 7.5 |
| 5 | 1 | 0.03 | 50 | 1 | -0.2 | 7.5 |
| SD v2.0 | | | | | | |
| 1 | 1 | 0.05 | 50 | 10 | -1.0 | 7.5 |
| 2 | 1 | 0.05 | 50 | 5 | -1.0 | 7.5 |
| 3 | 1 | 0.05 | 50 | 3 | -1.0 | 7.5 |
| 4 | 1 | 0.05 | 50 | 2 | -1.0 | 7.5 |
| 5 | 1 | 0.05 | 50 | 1 | -1.0 | 7.5 |

$\mathbf{x}_t$, as also demonstrated in Jeon et al. (2025). This means that when the model is conditioned on $t = 0$ and a memorized prompt, the signatures of memorization become prominent even if the input sample $\mathbf{x}_t$ is pure noise ($\mathbf{x}_T$). We empirically found that the high angular alignment (which is the signature of memorization in anisotropy) is present regardless of which denoised sample we use, i.e. one can ideally probe the model at $t \approx 0$ and use $x_{T \approx 0}$ (without intended mismatch) and could still detect memorization. However, this would require a denoising process which our method does not need. Though this is an intuitive explanation of the observed phenomenon, we believe that proving this intuition mathematically is outside the score of the paper and an interesting future work.

## A.5 LIMITATIONS AND FUTURE WORK

While our results demonstrate the superior performance and efficiency of the proposed method, there are some limitations to our approach. First, the optimal values of $\gamma_1$ and $\gamma_2$ used to report our results were determined specifically for SD v1.4 and SD v2.0. However, we empirically found that simple choices (e.g., $\gamma_1 = 1, \gamma_2 = 1$) generalize reasonably well across models with only minor performance degradation. To verify this, we compare our results using $\gamma$ values provided in Table 5 with the results when using arbitrary $\gamma$ values, specifically $\gamma_1 = 1$ and $\gamma_2 = 1$. The results are available in Table 7.

Table 7: Comparison of different values of $\gamma_1$ and $\gamma_2$ in our proposed metric. Here, $n$ represents the number of generations.

| Method | SD v1.4 | | SD v2.0 | |
|--------|---------|------------|---------|------------|
| | AUC ↑ | TPR@1%FPR ↑ | AUC ↑ | TPR@1%FPR ↑ |
| $n = 1$ | | | | |
| Original $\gamma$ values from Table 5 | 0.992 | 0.934 | 0.952 | 0.749 |
| $\gamma_1, \gamma_2 = 1$ | 0.990 | 0.918 | 0.949 | 0.721 |
| $n = 4$ | | | | |
| Original $\gamma$ values from Table 5 | 0.999 | 0.984 | 0.981 | 0.900 |
| $\gamma_1, \gamma_2 = 1$ | 0.997 | 0.974 | 0.979 | 0.868 |

We observe that the difference in performance between the two hyperparameter configurations is minimal. Hence, simple choices of $\gamma_1$ and $\gamma_2$ generalize across all models, and our approach does not heavily rely on the configurations of $\gamma_1$ and $\gamma_2$. When scaling to new settings, we recommend that one can always start with arbitrary $\gamma$ values to identify some memorized prompts and then later tune these weights to maximize the detection performance.

Another limitation of our work is that it does not focus on the distinction between local and global memorization. Specifically, the alignment component of our metric is less reliable in the cases of local memorization (as observed from Table 3). Therefore, developing a metric for local and global memorization that considers both anisotropy and isotropy, can be an interesting future research

direction. Lastly, like previous works, our evaluation only covers SD v1.4 and SD v2.0 due to the unavailability of memorized prompts data for other newer models, such as SD v3. Hence, future work could focus on identifying memorized prompts in newer large-scale models to enable more extensive evaluation.

## A.6  ADDITIONAL QUANTITATIVE RESULTS

To demonstrate the generalizability of our proposed metric beyond Stable Diffusion v1.4 and v2, we conduct our detection experiments on Realistic Vision v5.1 (CivitAI, 2023). We utilize the matching verbatim (MV) prompts from Webster (2023) for our experiments and compute our metric for 3 runs, each corresponding to a different seed. We report the AUC and TPR@1%FPR for number of generations $n = 1$ and $n = 4$, along with the time taken by our metric for 10 prompts in seconds in Table 8.

Table 8: Performance of our proposed metric on Realistic Vision v5.1. Here, $n$ represents the number of generations, and Time (sec.) represents the time taken for 10 metric calculations in seconds.

| Method | $n = 1$ | | | $n = 4$ | | |
|---|---|---|---|---|---|---|
| | AUC ↑ | TPR@1%FPR ↑ | Time (sec.) | AUC ↑ | TPR@1%FPR ↑ | Time (sec.) |
| $\mathcal{M}(\mathbf{x_T}, \mathbf{c})$ (ours) | $0.967 \pm 0.003$ | $0.778 \pm 0.002$ | 1.1 | $0.975 \pm 0.002$ | $0.756 \pm 0.004$ | 3.4 |

We observe that our method retains its detection capabilities in the Realistic Vision model, with a AUC above 0.96 and TPR@1%FPR above 0.75. Hence, these results demonstrate the generalization capabilities of our proposed metric beyond SD v1.4 and SD v2.0.

## A.7  ADDITIONAL QUALITATIVE RESULTS

We present additional visual results for our mitigation experiment in Figures 5 and 6. We qualitatively compare our approach with the methods from Ren et al. (2024) and Wen et al. (2024). We observe that our approach mitigates memorization more effectively than previous approaches by producing images of high quality that are well-aligned with the text prompt and distinct from the training image.

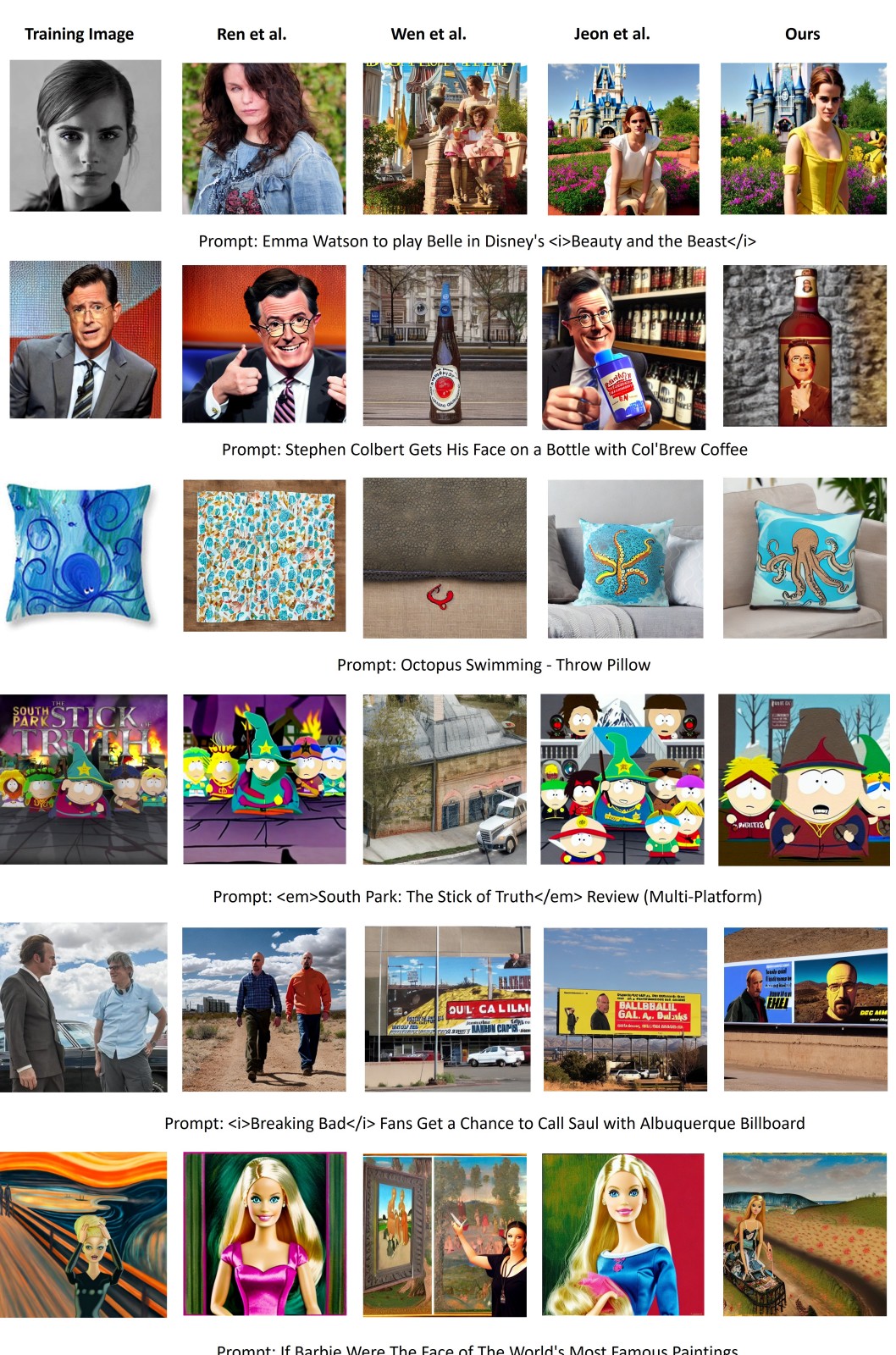

Figure 5: Qualitative comparison of inference-time mitigation approaches on SD v1.4. Specifically, we visualize the memorized training image (left-most), along with the mitigated generated images by the methods from Ren et al. (2024), Wen et al. (2024), Jeon et al. (2025), and our method (right-most).

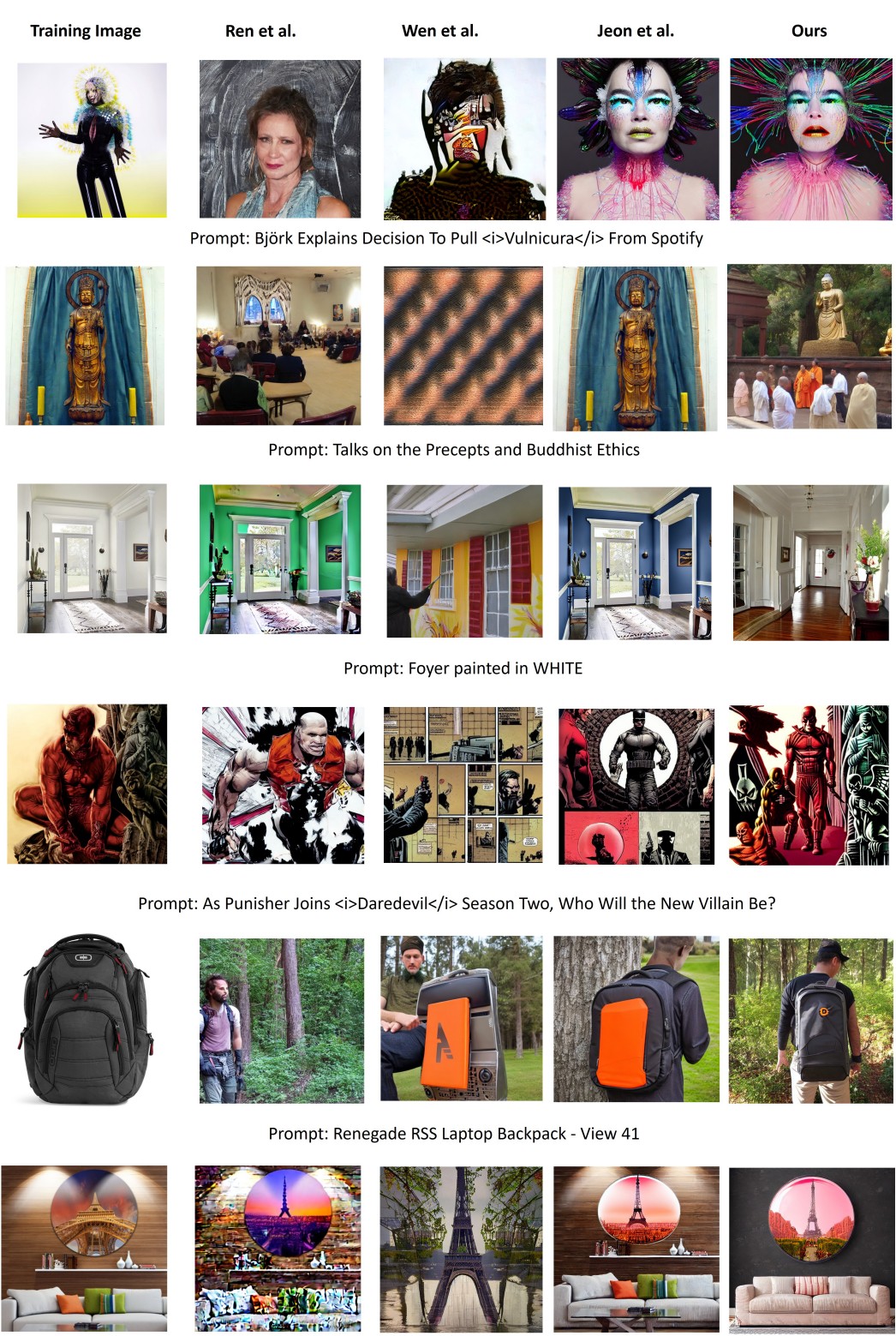

Figure 6: Qualitative comparison of inference-time mitigation strategies on SD v1.4. We show the memorized training image (left-most) alongside the mitigated generations by the approaches of Ren et al. (2024), Wen et al. (2024), Jeon et al. (2025), and our method (right-most).

