# OpenReview forum: "Detecting and Mitigating Memorization in Diffusion Models through Anisotropy of the Log-Probability"
_ICLR.cc/2026/Conference — ICLR 2026 Poster_

### Official Review · Reviewer_sxxs · 2025-10-29

**Soundness:** 2
**Presentation:** 2
**Contribution:** 2
**Rating:** 2
**Confidence:** 4

**Summary:**

The paper proposes a memorization detection (and mitigation) method for Stable Diffusion models that prioritizes angular alignment of scores over their norms, especially in the low-noise (late) phase of generation. It is positioned as a generalization of prior work (e.g., Wen 2024; Jeon 2025): earlier metrics largely emphasize the magnitude of a (conditional - unconditional) score difference, whereas this paper argues that direction (cosine similarity) carries the decisive signal under anisotropic noise. Empirically, the paper reports detection gains and mitigation impacts, but several theoretical claims and some experimental details need strengthening.

**Strengths:**

1. Conceptually extends prior works (Wen 2024; Jeon 2025) to anisotropic noise settings in diffusion models.
2. Provides initial empirical evidence suggesting angular features may correlate better with memorization risk.

**Weaknesses:**

1. The theoretical argument for “angular alignment is important” (Remark 1, Sec. 4.1) lacks rigor; Appendix A.1 remains heuristic without formal derivation or proof.
2. Confuses the norm of the score with the norm of the score difference; prior methods are not clearly distinguished.
3. The key computation $s(X_T, t\approx 0, c)$ is conceptually inconsistent, evaluating the final-time score on the initial noise requires justification or on-manifold correction.
4. Experimental details (Figure 2 setup, time-step sampling, model version, datasets) are missing, making reproduction difficult.
5. Comparison with Jeon 2025 mitigation results is absent, weakening empirical completeness.
6. Overall, the method’s novelty is somewhat limited without stronger theoretical grounding or broader validation.

**Questions:**

See weaknesses

---

> ### Author Response · Authors · 2025-11-22
> **Response to Reviewer sxxs (part 1)**
>
> We are thankful for the reviewer for their review and helpful comments. Below we address the main concerns of the reviewer.
>
> > The theoretical argument for “angular alignment is important” (Remark 1, Sec. 4.1) lacks rigor; Appendix A.1 remains heuristic without formal derivation or proof.
>
> We thank the reviewer for this suggestion focused on improving the mathematical rigor. **Upon the reviewer's request, we have now included a formal theorem regarding alignment in the main text (Section 4.2), along with a mathematical proof of the theorem in the Appendix Section A.1.** The Theorem and Proof added are as follows:
>
> #### Theorem 1:
> Consider the anisotropic low-noise regime of diffusion and let $\Sigma\_t,\Sigma^c\_t$ be symmetric positive definite and $v\_t := \mathbf{x}\_t-\mu$ and $\delta := \mu\_c-\mu$ denote the sample displacement from the unconditional mode and the relative displacement between the guidance mode and unconditional mode. Then,
>
> $s\_\theta(\mathbf{x}\_t, t) = \nabla\_{\mathbf{x\_t}}\log p\_t(\mathbf{x}\_t) := -\Sigma^{-1}\_tv\_t;\quad$
> $s\_\theta^\Delta(\mathbf{x}\_t, t ,c)= \nabla\_{\mathbf{x\_t}}\log p\_t(c|\mathbf{x}\_t):= (\Sigma^{-1}\_t-\Sigma\_t^{c^{-1}})v\_t + \Sigma\_t^{c^{-1}}\delta.$
>
> Let $\alpha>0$ and constants $\varepsilon,\tau \ge 0$.
>
> Assume $\|(\Sigma^{-1}\_t-\Sigma\_t^{c^{-1}})v\_t - \alpha s\_\theta(\mathbf{x}\_t, t)\|\le \varepsilon \alpha \|s\_\theta(\mathbf{x}\_t, t)\|,\qquad
> \|\Sigma\_t^{c^{-1}}\delta\|\le \tau \alpha \|s\_\theta(\mathbf{x}\_t, t)\|,$
> and set $r:=\varepsilon+\tau<1$.
>
> Then the cosine similarity satisfies
> $\cos(s\_\theta(\mathbf{x}\_t, t),s\_\theta^\Delta(\mathbf{x}\_t, t ,c)) \ge\ \frac{1-r}{1+r}.$
>
> This theorem provides a lower-bound for the cosine similarity between $\nabla\_{\mathbf{x}\_t}\log p\_t(\mathbf{x}\_t)$ and $\nabla\_{\mathbf{x}\_t}\log p\_t(c | \mathbf{x}\_t)$ using the relative displacement ($\delta$) between the guidance mode and the unconditional mode. Specifically, if both the unconditional and guidance modes nearly coincide ($\delta \rightarrow 0$), the only deviation between $s\_\theta^\Delta(\mathbf{x}\_t, t, c)$ and scaled $s\_\theta(\mathbf{x}\_t, t)$ arises from differences in covariance, which are controlled by the approximation error $\varepsilon$. The resulting error term $r = \varepsilon + \tau$ simplifies to $r = \varepsilon$, and the cosine similarity lower bound in the theorem becomes $\frac{1-\varepsilon}{1+\varepsilon}$. Thus, if $\varepsilon$ is small, the alignment between $\nabla\_{\mathbf{x}\_t}\log p\_t(\mathbf{x}\_t)$ and $\nabla\_{\mathbf{x}\_t}\log p\_t(c | \mathbf{x}\_t)$ becomes high.
> #### Proof:
> By definition,
>
> $s\_\theta^\Delta(\mathbf{x}\_t, t ,c) \;=\; (\Sigma\_t^{-1}-\Sigma\_t^{c^{-1}})v\_t + \Sigma\_t^{c^{-1}}\delta.$
>
> By assumption, there exists $\alpha>0$ such that
> $(\Sigma\_t^{-1}-\Sigma\_t^{c^{-1}})v\_t= \alpha s\_\theta(\mathbf{x}\_t, t)  + \Delta\_1,\qquad
> \|\Delta\_1\|\le \varepsilon \alpha \|s\_\theta(\mathbf{x}\_t, t) \|,$
>
> and
> $\Sigma\_t^{c^{-1}}\delta = \Delta\_2,\qquad
> \|\Delta\_2\|\le \tau \alpha \|s\_\theta(\mathbf{x}\_t, t)\|.$
>
> Thus,
> $s\_c = \alpha s\_\theta(\mathbf{x}\_t, t) + \Delta\_1 + \Delta\_2 = \alpha s\_\theta(\mathbf{x}\_t, t) + \Delta,$
>
> where $\Delta = \Delta\_1+\Delta\_2$ satisfies
> $\|\Delta\| \le (\varepsilon+\tau)\alpha \|s\_\theta(\mathbf{x}\_t, t)\| = r\alpha \|s\_\theta(\mathbf{x}\_t, t)\|.$
>
> The cosine similarity is
> $\cos(s\_\theta(\mathbf{x}\_t, t),s\_\theta^\Delta(\mathbf{x}\_t, t ,c)) =
> \frac{\langle s\_\theta(\mathbf{x}\_t, t), \alpha s\_\theta(\mathbf{x}\_t, t) + \Delta \rangle}{\|s\_\theta(\mathbf{x}\_t, t)\|\,\|\alpha s\_\theta(\mathbf{x}\_t, t) + \Delta\|}.$
>
> For the numerator,
> \begin{equation}
> \begin{aligned}
> \langle s\_\theta(\mathbf{x}\_t, t), \alpha s\_\theta(\mathbf{x}\_t, t) + \Delta \rangle
> &= \alpha \|s\_\theta(\mathbf{x}\_t, t)\|^2 + \langle s\_\theta(\mathbf{x}\_t, t),\Delta\rangle \\
> &\ge\; \alpha \|s\_\theta(\mathbf{x}\_t, t)\|^2 - \|s\_\theta(\mathbf{x}\_t, t)\|\|\Delta\| \\
> &\ge\; (1-r)\alpha \|s\_\theta(\mathbf{x}\_t, t)\|^2.
> \end{aligned}
> \end{equation}
>
> For the denominator,
> $\|\alpha s\_\theta(\mathbf{x}\_t, t) + \Delta\| \;\le\; \alpha \|s\_\theta(\mathbf{x}\_t, t)\| + \|\Delta\|
> \;\le\; (1+r)\alpha \|s\_\theta(\mathbf{x}\_t, t)\|.$
>
> Therefore,
> $\cos(s\_\theta(\mathbf{x}\_t, t),s\_\theta^\Delta(\mathbf{x}\_t, t ,c)) \;\ge\; \frac{(1-r)\alpha \|s\_\theta(\mathbf{x}\_t, t)\|^2}{\|s\_\theta(\mathbf{x}\_t, t)\|(1+r)\alpha \|s\_\theta(\mathbf{x}\_t, t)\|}
> = \frac{1-r}{1+r}.$
>
> This completes the proof.
>
> > Confuses the norm of the score with the norm of the score difference; prior methods are not clearly distinguished.
>
> We thank the reviewer for pointing this out and we understand how these two terminologies could have been confusing. **To improve clarity and consistency of terminologies, we have replaced the term "norm of the score estimates" with "norm of the score difference" wherever possible in our updated paper**.

---

> ### Author Response · Authors · 2025-11-22
> **Response to Reviewer sxxs (part 2)**
>
> > The key computation $s(x_T, t \approx0,c)$ is conceptually inconsistent, evaluating the final-time score on the initial noise requires justification or on-manifold correction.
>
> We appreciate the reviewer for this thoughtful criticism. We argue that one plausible explanation on why probing a model at $t\approx0$ while inputing high noise works so well in memorization detection is **because memorization is encoded in the learned log-probability of the trained model and is mostly independent of the input sample $\mathbf{x}\_t$, as also demonstrated in [1]**. This means that when the model is conditioned on $t\approx0$ and a memorized prompt, the signatures of memorization become prominent even if the input sample $\mathbf{x}_t$ is pure noise ($\mathbf{x}\_T$). Therefore, the high angular alignment (which is the signature of memorization in anisotropy) is present regardless of which denoised sample we use, i.e. one can ideally probe the model at $t\approx0$ and use $x\_{T\approx0}$ (without intended mismatch) and could still detect memorization. However, this would require a denoising process which our method does not need. Though this is an intuitive explanation of the observed phenomenon, we believe that proving this intuition mathematically is outside the scope of this paper and an interesting future direction. **We have included this explanation in the Appendix Section A.4 of our updated manuscript.**
>
> > Experimental details (Figure 2 setup, time-step sampling, model version, datasets) are missing, making reproduction difficult.
>
> **To facilitate reproduction of our method and results, we have now added our code to the supplementary material of this submission. We will additionally upload and maintain a gitub repository later.** Moreover, we have included experimental details for our Figure 2 experiment in the Appendix Section A.3.1 of our updated paper. Other experimental details are already provided in Section A.3.
>
> > Comparison with Jeon 2025 mitigation results is absent, weakening empirical completeness.
>
> We thank the reviewer for raising this point. Since Jeon et. al. (2025) [1] does not utilize a prompt-based mitigation method like [2] and [3], the numbers reported in [1] are not directly comparable with ours, and hence were omitted in the original submission. **We have now implemented the metric from Jeon et al. [1] in our prompt-based mitigation method, and reported the quantitative and qualtitative results in Figures 4,5, and 6 of our updated paper.** We observe that Jeon et al. is on par with our approach in SD v1.4 and slightly outperforms our approach on SD v2.0 but is also at least 5x slower, as also demonstrated in the detection experiments.
>
> ### References
> [1] Jeon, Dongjae, Dueun Kim, and Albert No. "Understanding Memorization in Generative Models via Sharpness in Probability Landscapes." arXiv preprint arXiv:2412.04140 (2024).
>
> [2] Wen, Y., Liu, Y., Chen, C., & Lyu, L. (2024). Detecting, Explaining, and Mitigating Memorization in Diffusion Models. The Twelfth International Conference on Learning Representations.
>
> [3] Brendan Leigh Ross, Hamidreza Kamkari, Tongzi Wu, Rasa Hosseinzadeh, Zhaoyan Liu, George Stein, Jesse C. Cresswell, and Gabriel Loaiza-Ganem. A geometric framework for understanding memorization in generative models. In The Thirteenth International Conference on Learning Representations, 2025

---

### Official Review · Reviewer_Kb53 · 2025-10-30

**Soundness:** 3
**Presentation:** 3
**Contribution:** 2
**Rating:** 4
**Confidence:** 4

**Summary:**

The paper studies memorization in diffusion models through the lens of anisotropy in the log-probability landscape. It argues that prior norm-based metrics are only effective under isotropy and introduces a denoising-free detection metric that combines (i) cosine alignment between conditional and unconditional scores in the anisotropic regime and (ii) score-norm magnitude in the isotropic regime. The method requires only two forward passes—conditional and unconditional—and is shown to outperform existing denoising-free metrics on Stable Diffusion v1.4 and v2.0 while being significantly faster.

**Strengths:**

•	The identification of anisotropy as a key factor in memorization detection is conceptually strong and empirically supported (variance of Hessian eigenvalues increasing toward low noise).

•	The method is denoising-free, computationally efficient, and integrates both magnitude and directional cues.

•	Experiments follow prior evaluation standards and report competitive or superior results. Using new designated bencharks such as MemBench is also a bonus.

**Weaknesses:**

•	The section titled “Failure of Norm-Based Methods in Anisotropy”	 starts with “We now prove…” However it does not constitute a formal proof. Rephrase to “demonstrate” or “show analytically” or similar.



•	In Table 1, please report standard deviations or confidence intervals. With 500 prompts, small improvements (such as the +0.001 AUC over Jeon et al. at n = 4, SD v1.4) may fall within statistical noise.



•	Fig. 3 (a), (b) should be key visualizations supporting the main hypothesis - however they are insufficiently explained.



•	Qualitative examples of generations with mitigated memorization are absent. Please provide such a comparison with the competitors. In addition, it is preferable to add an exhaustive set of such generations to the appendix.



•	Please refer to the concurrent [1], which employs a different criterion that similarly emphasizes direction over magnitude. The numerical computation of its curvature criterion sums angles (cosines) between the conditioning score and surface normals. Clarifying how this differs from your approach, and whether it fits within your anisotropic framework, would aid contextualization for future reference.



•	Please acknowledge [2] in the related work. The claim that “this phenomenon was firstly identified by Somepalli et al. (2023a)” overlooks [2], which concurrently with Somepalli et. al, demonstrated training data extraction (memorized samples) from diffusion models.



•	The inference-time mitigation description lacks clarity - reducing reproducibility. Please specify how γ₁ and γ₂ were selected and how the score at ($t \approx 0$) is obtained without full denoising (a clarification in the Appendix would suffice).



-----



[1] Brokman et al. (2025). Tracking Memorization Geometry throughout the Diffusion Model Generative Process. NeurIPS 2025 Workshop on Symmetry and Geometry in Neural Representations.



Link: https://openreview.net/forum?id=4XSVk26sHj



[2] Carlini et al. (2023). Extracting Training Data from Diffusion Models. USENIX Security Symposium.



Link: https://www.usenix.org/conference/usenixsecurity23/presentation/carlini

**Questions:**

Please see weaknesses above.

---

> ### Author Response · Authors · 2025-11-22
> **Response to Reviewer Kb53 (part 1)**
>
> We extend our sincere appreciation to the reviewer for their valuable suggestions and insightful comments. Below, we address the concerns of the review.
>
>
> > The section titled “Failure of Norm-Based Methods in Anisotropy” starts with “We now prove…” However it does not constitute a formal proof. Rephrase to “demonstrate” or “show analytically” or similar.
>
> We thank the reviewer for the suggestion. **We have changed our wording in Section 4.1 from "prove" to "demonstrate" in the updated manuscript.**
>
> > In Table 1, please report standard deviations or confidence intervals. With 500 prompts, small improvements (such as the +0.001 AUC over Jeon et al. at n = 4, SD v1.4) may fall within statistical noise.
>
> We thank the reviewer for this helpful comment. **We have now conducted our detection experiment for 3 runs, each with a different seed, and reported the mean and standard deviation for the runs.** The results are as follows.
>
>
> | Method                               | AUC               | TPR@1%FPR   | Time (sec.)       | AUC         | TPR@1%FPR   | Time (sec.) |
> | ------------------------------------ | ----------------- | ----------- | ----------------- | ----------- | ----------- | ----------- |
> |                                      | SD v1.4  | SDv1.4  |  SDv1.4           | SDv2.0            | SDv2.0|  SDv2.0                                  |                   |             |                   |             |             |             |
> |                                  |                  |              |     $\textbf{n=1}$              |             |             |             |
> | Ren et al. (2024)                    | 0.846             | 0.116       | 0.05              | 0.848       | 0           | 0.07        |
> | Wen et al. (2024)                    | 0.976             | 0.896       | 0.40              | 0.948       | 0.739       | 0.80        |
> | Jeon et al. (2025)                   | 0.987             | 0.908       | 5.40              | $\textbf{0.959}$       | 0.740       | 14.60       |
> | $\mathcal{M}(\mathbf{x}_T,c)$ (ours) | **0.994 ±0.001**      | **0.935±0.002** | 1.10              | $\underline{0.953±0.016}$ | **0.791±0.015** | 2.20        |
> |                                      |                   |             |                   |             |             |             |
> |                                   |                   |             |                 $\textbf{n=4}$   |             |            |             |
> | Ren et al. (2024)                    | 0.839             | 0.130       | 0.05              | 0.853       | 0           | 0.07        |
> | Wen et al. (2024)                    | 0.992             | 0.944       | 1.20              | 0.980       | 0.876       | 2.70        |
> | Jeon et al. (2025)                   | 0.998             | 0.982       | 19.40             | **0.991**       | **0.895**       | 56.40       |
> | $\mathcal{M}(\mathbf{x}_T,c)$ (ours) | **0.999±0.001**       | **0.984±0.002** | 3.40              | $\underline{0.981±0.003}$ | $\underline{0.890±0.010}$ | 7.30        |
>
> We observe from the StD results on SD v1.4 that our method is highly stable and not sensitive to random seeds with all the deviations $\leq0.002$ . Moreover, we observe that the StD values of TPR@1\%FPR are generally higher than AUC, suggesting that TPR@1\%FPR is more sensitive to random seed. **We have included these extended results in Table 1 of our updated manuscript.**
>
>
>
> > Fig. 3 (a), (b) should be key visualizations supporting the main hypothesis - however they are insufficiently explained.
>
> **We have now further elaborated our results from the Figure 3 experiment and added some experimental details for the same.** In specific, we observe from the results that memorized samples exhibit significantly higher alignment, and thus, intuitively, higher cosine similarity between the conditioning term and the unconditional score estimates. This means that for the memorized cases in anisotropy, conditioning does not introduce new directions to the unconditional gradient. In contrast, non-memorized cases generally exhibit misaligned directions, implying weak or random angular correlation between $\nabla_{\mathbf{x}_t}\log p_t(\mathbf{x}_t)$ and $\nabla\_{\mathbf{x}\_t}\log p\_t(c | \mathbf{x}\_t)$. This behavior aligns with our theoretical prediction: in anisotropic neighborhoods surrounding memorized data, the conditional log-probability mode coincides with the mode of unconditional log-probability, producing high angular alignment.

---

> ### Author Response · Authors · 2025-11-22
> **Response to Reviewer Kb53 (part 2)**
>
> > Qualitative examples of generations with mitigated memorization are absent. Please provide such a comparison with the competitors. In addition, it is preferable to add an exhaustive set of such generations to the appendix.
>
> We thank the reviewer for this valuable suggestion. Qualitative comparisons of generations after mitigation, including those from competing methods, were already provided in the Appendix (Figures 5, 6) of the original manuscript. **To improve visibility and accessibility, we have now moved a representative subset of these examples from the Appendix to the main text (Figure 4) in our updated manuscript.** These examples illustrate that our mitigation strategy effectively mitigates memorization while preserving the prompt-image alignment. The full set of qualitative results remains available in the Appendix for completeness.
>
> > Please refer to the concurrent [1], which employs a different criterion that similarly emphasizes direction over magnitude. The numerical computation of its curvature criterion sums angles (cosines) between the conditioning score and surface normals. Clarifying how this differs from your approach, and whether it fits within your anisotropic framework, would aid contextualization for future reference.
>
> We thank the reviewer for pointing out this concurrent work, **which we now cite and discuss in the revised manuscript (Section 2, Related Work).** While the approach from [1] tracks how local curvature evolves throughout the denoising trajectory to characterize memorization, our method focuses on angular alignment between the conditional and unconditional scores in the low-noise anisotropic regime. Despite these methodological differences, both approaches share the common intuition that memorization is encoded in the directional geometry of the learned score field rather than in its magnitude. Thus, we note that [1] can be seen as a higher-order extension of our framework.
>
> > Please acknowledge [2] in the related work. The claim that “this phenomenon was firstly identified by Somepalli et al. (2023a)” overlooks [2], which concurrently with Somepalli et. al, demonstrated training data extraction (memorized samples) from diffusion models.
>
> Thank you for the suggestion. **We have now included [2] in our updated related work (Section 2).**
>
> > The inference-time mitigation description lacks clarity - reducing reproducibility. Please specify how $\gamma_1$ and $\gamma_2$ were selected and how the score at $(t\approx0)$ is obtained without full denoising (a clarification in the Appendix would suffice).
>
> We thank the reviewer for this helpful comment. **To enhance reproducability, we have now uploaded the code as a Supplementary Material to this submission.  We will additionally upload and maintain a gitub repository upon acceptance.** As described in Appendix Section A.3 of the original manuscript, the hyperparameters $\gamma_1$ and $\gamma_2$ were chosen by fitting a simple Logistic Regressor once on a small set (of size 20) of memorized prompts. Then, we use the found optimal values for all of our detection and mitigation experiments. We also show that our results are not heavily reliant on these hyperparameters with arbitrary hyperparameter configurations resulting in minimal performance loss (Table 5, Section A.5). Lastly, explanation on how the score at $(t\approx0)$ is obtained without full denoising is detailed in Section 4.3 of the original manuscript. We calculate our metric at artificially set timesteps $t=1$ and $t=T$ using the same initial Gaussian noise sample $\mathbf{x}_T \sim \mathcal{N}(\mathbf{0,I})$, i.e., we do not run a reverse denoising trajectory and simply condition the model on different timesteps $t=1$ and $t=T$.
>
> ### References
>
> [1] Brokman et al. (2025). Tracking Memorization Geometry throughout the Diffusion Model Generative Process. NeurIPS 2025 Workshop on Symmetry and Geometry in Neural Representations.
>
> [2] Carlini et al. (2023). Extracting Training Data from Diffusion Models. USENIX Security Symposium.

---

> > ### Comment · Reviewer_Kb53 · 2025-11-24
> >
> > Thank you to the authors for their efforts in improving the manuscript. My concerns have been satisfactorily addressed, and I would raise my score accordingly.

---

### Official Review · Reviewer_y8dV · 2025-10-31

**Soundness:** 3
**Presentation:** 3
**Contribution:** 3
**Rating:** 6
**Confidence:** 3

**Summary:**

The paper investigates memorization in text-to-image diffusion models and argues that existing norm-based detection methods only work under isotropic log-probability assumptions, which is the case for high or medium noise levels. It formalizes how anisotropy arises in the low-noise regime and proposes a new denoising-free metric that is defined as the cosine similarity between conditional and unconditional score estimates. This new metric is combined with the known “isotropic term” - norm of the guidance vector. The method requires only two denoising steps (only noise estimation, not real denoising). The method improves AUC and TPR (at 1% FPR) on Stable Diffusion v1.4/v2.0. It also integrates the metric into an inference-time prompt-optimization scheme for mitigation, evaluated on MemBench.

**Strengths:**

1.	Sound theoretical framing. The analysis clearly connects memorization signatures to the isotropy and anisotropy regimes of the log-probability, filling a conceptual gap in prior isotropic norm-based methods.
2.	Efficient and simple metric. The proposed anisotropy-aware score is computationally cheap (two model steps, no actual denoising) yet achieves higher detection accuracy.
3.	The method is not limited to detection, it also demonstrates practical mitigation by optimizing prompt embeddings, producing non-memorized images while maintaining text alignment and aesthetic quality.

**Weaknesses:**

1. The reported “speed-up” is only shown relative to one denoising-free baseline, while simpler existing methods remain faster with no meaningful loss in performance, limiting the practical efficiency benefit of the proposed approach.
2. Incremental improvement weakness: AUC/TPR gains are marginal because prior methods already near saturation, making the contribution incremental rather than substantively advancing the state of the art, even if the method itself is technically sound.
3. No code to allow future comparison, and no indication that it will be provided in the future (Wen et al., which the authors compare to, do provide code). I urge the authors to add code so that their work will be maximally helpful to others.
4. The imporovements are small. For mitigation the improvement looks more consistent, and for detection, Wen et al. are (~3x) faster and slighty worse, while Jeon et al. are on par and much slower (~6x).
5. Minor - Equation 8 typo: the unconditional score in the cosine similarity term should not have “c” in it.

**Questions:**

1.	Given that prior methods already achieve near-saturated AUC/TPR, how do you justify the significance of the contribution beyond small numerical gains (with x2-3 in latency)?
2.	Since your claimed speed-up is only relative to a single denoising-free baseline, and simpler non-denoising-free methods remain faster with comparable performance (Wen et al.), can you clarify in what scenarios your method is actually the preferred choice in practice?
3.	Regarding the t=0 probe using high-noise latents - the model is inputted a high-noise latent but set with t=0, meaning the model “believes” it is operating at the end of the denoising process, which the authors showed is the low-noise regime. However the actual noise latent is far from the data manifold. Can you explain why a cosine similarity computed under this deliberate mismatch, based on what the model thinks rather than the actual noise level, should remain reliable and theoretically meaningful for detecting memorization (as empirically demonstrated)?
4.	Ablation - table 3 (in the Appendix) shows that gamma_1 differs is 2 and 0.1in  SD v1.4 and SD v2.0, suggesting that the anisotropy term is unstable and requires model-specific tuning.  Can you explain why such tuning is necessary, and can you provide an ablation where we can understand the individual contribution and necessity of each component (gamma1 =0  and gamma2=0 in separate settings)?

---

> ### Author Response · Authors · 2025-11-22
> **Response to Reviewer y8dV (part 1)**
>
> We are sincerely thankful to the reviewer for the insightful comments and valuable suggestions to our paper. Below, we address the reviewer's main concerns:
>
> > The reported “speed-up” is only shown relative to one denoising-free baseline, while simpler existing methods remain faster with no meaningful loss in performance, limiting the practical efficiency benefit of the proposed approach.
>
> We appreciate the reviewer’s concern and agree that Wen et al. is computationally faster. While our method is on average 0.2 seconds slower (per prompt) than Wen et al., we would like to point out that our method thrives at borderline cases, where Wen et al. shows under-detection of memorization. This can be seen in our qualitative results (Figures 4,5,6) and the quantitative results (Table 1), where under stict-false positive constraints, our method shows improvements of TPR@1%FPR by upto 5.1% (average improvement of 3.6%) compared to Wen et. al. This is because our method combines independent predictors of memorization and thus, is more robust to these edge-cases. We argue that this increment is critical, especially in scenarios where minimizing false-negatives is more important than compromising the detection speed by ~0.2 seconds. **We have included this explanation in our updated manuscript (Section 5.1).**
>
>
> > Since your claimed speed-up is only relative to a single denoising-free baseline, and simpler non-denoising-free methods remain faster with comparable performance (Wen et al.), can you clarify in what scenarios your method is actually the preferred choice in practice?
>
> The ability of our method to work in the anisotropic regime have several practical benefits, mainly considering the scenarios where the access to anisotropic diffusion regime is easier/more efficient compared to the isotropic regime. For example, in image level memorization task [1], the aim is to detect memorized images (or parts of it) only through the access of generated images. These generated images are far from the high-noise isotropic regime and are at the data manifold. A typical approach [1] is to use a DDIM inversion process to get a noisy latent, and then perform forward processes on perturbed latents. This requires the DDIM inverter to go back to initial noise timesteps and then perform denoising through a large number of time steps, if we use Wen's (isotropic) metric. Thus, utilizing a metric which works under the anisotropic regime should be preferable in this case as one would only require to invert and denoise a few time steps to compute our metric. **We have included this explanation in the introduction (Section 1) of our updated paper**.
>
> > No code to allow future comparison, and no indication that it will be provided in the future (Wen et al., which the authors compare to, do provide code). I urge the authors to add code so that their work will be maximally helpful to others.
>
> We thank the reviewer for this suggestion. We were planning to upload and maintain a gitub repository upon acceptance to facilitate reproducability but mistakenly did not explicitly mention that in our paper. **We have now also provided the code as Supplementary Material in this submission.**
>
> > Minor - Equation 8 typo: the unconditional score in the cosine similarity term should not have “c” in it.
>
> We appreciate the reviewer for noticing the typo. **It is now fixed in our updated manuscript.**
>
>
> > Given that prior methods already achieve near-saturated AUC/TPR, how do you justify the significance of the contribution beyond small numerical gains (with x2-3 in latency)?
>
> As the reviewer correctly mentioned, the AUC/TPR values are nearly saturated and it is practically impossible to beat those methods by a large margin. However, we argue that understanding and rectifying the limitations of previous memorization detection approaches is still an important research direction for mitigating safety, privacy, and copyright issues in large scale diffusion models. The contribution of our work, beyond numerical gains, is a deeper understanding of the failure of recent approaches when working in the anisotropic regime of diffusion. This is critical in scenarios where the access to isotropic diffusion regime is harder to obtain than the anisotropic regime, e.g. in image-level memorization tasks.

---

> ### Author Response · Authors · 2025-11-22
> **Response to Reviewer y8dV (part 2)**
>
> > Regarding the t=0 probe using high-noise latents...Can you explain why a cosine similarity computed under this deliberate mismatch should remain reliable and theoretically meaningful for detecting memorization?
>
> We thank the reviewer for this very insightful question. We argue that one plausible explanation on why probing a model at $t\approx0$ while inputing high noise works so well in memorization detection is **because memorization is encoded in the learned log-probability of the trained model and is mostly independent of the input sample $\mathbf{x}\_t$, as also demonstrated in [2]**. This means that when the model is conditioned on $t\approx0$ and a memorized prompt, the signatures of memorization become prominent even if the input sample $\mathbf{x}\_t$ is pure noise ($\mathbf{x}\_T$). Therefore, high angular alignment (which is the signature of memorization in anisotropy) is present regardless of which denoised sample we use, i.e. one can ideally probe the model at $t\approx0$ and use $x_{T\approx0}$ (without intended mismatch) and could still detect memorization. However, this would require a denoising process which our method does not need. Though this is an intuitive explanation of the observed phenomenon, we believe that proving this intuition mathematically is outside the scope of this paper and an interesting future direction. **We have included this explanation in the Appendix Section A.4 of our updated manuscript.**
>
> > Can you explain $\gamma$ tuning is necessary?
>
> Thank you for the question. We would like to point out that as shown in Appendix Section A.5 (A.4 of the original submission), our method does not heavily rely on the $\gamma$ tuning and the untuned metric (with arbitrary $\gamma$ values $\gamma_1=1, \gamma_2=1$) performs almost as good as the tuned one. The sole purpose of the tuning is to leverage the maximum detection performance but in realistic settings, arbitrary $\gamma$ values work quite well.
>
> > Can you provide an ablation where we can understand the individual contribution and necessity of each component (gamma1 =0 and gamma2=0 in separate settings)?
>
> We thank the reviewer for the suggestion for the ablation. **Upon reviewer's request, we conducted an ablation study, analzing the contribution of each component**. The results are presented below.
>
> | Method                                  | AUC $\uparrow$  | TPR@1%FPR $\uparrow$ | AUC $\uparrow$  | TPR@1%FPR $\uparrow$ |
> |------------------------------------------|--------|-------------|--------|-------------|
> |               | **SDv1.4** | **SDv1.4** | **SDv2.0** | **SDv2.0**|
> |                                  |         |              |        |             |
> |                       |        |        **$\mathbf{n=1}$**       |        |             |
> | Norm of the score difference in isotropy | 0.976  | 0.896       | 0.948  | 0.739       |
> | Cosine similarity in anisotropy          | 0.923  | 0.424       | 0.779  | 0.416       |
> | **Combined (ours)**                      | **0.992**  | **0.934**       | **0.952**  | **0.749**       |
> |                       |        |       **$\mathbf{n=4}$**        |        |             |
> | Norm of the score difference in isotropy | 0.992  | 0.944       | 0.980  | 0.876       |
> | Cosine similarity in anisotropy          | 0.939  | 0.440       | 0.785  | 0.401       |
> | **Combined (ours)**                      | **0.999**  | **0.984**       | **0.981**  | **0.900**       |
>
> We observe that the norm of the score difference generally performs better than cosine similarity. However, the combination of the two terms exceeds the performance of both terms individually. Specifically, we find that the cosine similarity performs worse in SD v2.0 compared to SD v1.4. This is because the memorized prompt set for SD v2.0 consists mostly of local memorization cases, where only some parts of training set are memorized. In these cases, the mode displacement $\delta$ between $\log p_t(\mathbf{x}_t)$ and $\log p_t(c|\mathbf{x}_t)$ is large because the other non-memorized features increase the mode distance. Hence, the cosine similarity becomes lower, and the alignment is no longer a reliable metric. Therefore, the combination of these two metrics is necessary for robust local memorization detection and mitigation. **We have added this ablation (Section A2.2) and added an explanation regarding the limitations of the alignment component of our metric in Section A.5.**
>
> ### References
>  [1] Jiang, Y., Lin, H., Bai, Y., Peng, B., Liu, Z., Lyu, Y., … Dong, J. (2025). Image-level Memorization Detection via Inversion-based Inference Perturbation. The Thirteenth International Conference on Learning Representations.
>
>  [2] Jeon, Dongjae, Dueun Kim, and Albert No. "Understanding Memorization in Generative Models via Sharpness in Probability Landscapes." arXiv preprint arXiv:2412.04140 (2024).

---

### Official Review · Reviewer_WjxY · 2025-11-01

**Soundness:** 3
**Presentation:** 2
**Contribution:** 3
**Rating:** 6
**Confidence:** 4

**Summary:**

This paper addresses memorization in text-to-image diffusion models, a phenomenon where models reproduce training samples. The authors argue that existing norm-based memorization detection methods, which rely on the magnitude of score estimates, implicitly assume isotropic log-probability distributions, an assumption that holds mainly in high- or mid-noise regimes. The paper identifies that in low-noise (anisotropic) regimes, memorization manifests instead through strong angular alignment between conditional and unconditional score functions. Based on this insight, the authors propose a new denoising-free memorization detection metric that combines (1) the cosine similarity between guidance and unconditional score vectors in the anisotropic regime and (2) the score-norm difference in the isotropic regime. This metric requires only two forward passes (conditional and unconditional), making it substantially faster than prior methods. Experiments on Stable Diffusion v1.4 and v2.0 show improved AUC and TPR@1%FPR compared to previous denoising-free baselines, with up to 5× speedup. The method is also used for inference-time mitigation via prompt augmentation, demonstrating improved trade-offs between similarity, CLIP, and aesthetic scores on MemBench.

**Strengths:**

1. The paper is well-organized, making it easy to follow. The visualizations clearly illustrate the messages and insights it wishes to deliver.
2. This paper contributes to privacy-preserving text-to-image diffusion models, which are practically significant for preventing copyright risks.
3. The proposed metric’s efficiency (denoising-free and fast) makes it appealing for large-scale model auditing.
4. The paper introduces a new theoretical and empirical perspective on diffusion model memorization by analyzing the anisotropy of log-probability. The conceptual shift from norm magnitude to angular alignment in the low-noise regime is novel and well-motivated. The derived connection between isotropy, anisotropy, and curvature sharpness is mathematically supported and empirically verified.
5. Experimental results are comprehensive, covering multiple Stable Diffusion versions, standard benchmarks, and ablations. Comparisons include strong baselines and standard metrics.

**Weaknesses:**

1. The paper could benefit from additional discussion on whether the proposed metric generalizes across architectures beyond SD v1.4/v2.0.
2. Although Appendix A.4 shows some robustness, the weights are empirically tuned per model, suggesting potential calibration issues when scaling to new settings.
3. The proposed cosine-similarity measure is intuitive but may be sensitive to normalization choices. A sensitivity analysis on score normalization or noise schedule parameters would strengthen the robustness claims.
4. Despite being well-organized, there are several equations in the paper that remain unlabelled on page 5 (between equations 5 and 6).
5. The discussion on Figure 2’s overlap could benefit from quantitative metrics (e.g., KL divergence between distributions).

**Questions:**

See the above strengths and weaknesses.

---

> ### Author Response · Authors · 2025-11-22
> **Response to Reviewer WjxY**
>
> We would like to express our sincere gratitude to the reviewer for the comprehensive and detailed review, which helped us further improve the quality of our paper. To address the reviewer's concerns, below we provide clarification on each point raised in a sequential manner.
>
> > The paper could benefit from additional discussion on whether the proposed metric generalizes across architectures beyond SD v1.4/v2.0.
>
> We thank the reviewer for their helpful suggestion. **To demonstrate the generalization of our proposed metric, we have now conducted additional detection experiments on one of the latest Realistic Vision v5.1 model.** The results are presented below.
>
> | Method                                | AUC ↑ (n=1) | TPR@1%FPR ↑ (n=1) | Time (sec.) (n=1) | AUC ↑ (n=4) | TPR@1%FPR ↑ (n=4) | Time (sec.) (n=4) |
> | ------------------------------------- | ----------- | ----------------- | ----------------- | ----------- | ----------------- | ----------------- |
> | $\mathcal{M}(\mathbf{x}_T, c)$ (ours) | 0.967 $\pm$ 0.003       | 0.778 $\pm$ 0.002             | 1.1               | 0.975 $\pm$ 0.002      | 0.756 $\pm$ 0.004             | 3.4               |
>
> We observe that our method retains its detection capabilities in the Realistic Vision model, with a AUC above 0.96 and TPR@1\%FPR above 0.75. Hence, these results demonstrate the generalization capabilities of our proposed metric beyond SD v1.0 and SD v2.0. **We have included the detection experiment on Realistic Vision v5.1 in the Appendix Section A.6 of our updated manuscript.**
>
> > Although Appendix A.4 shows some robustness, the weights are empirically tuned per model, suggesting potential calibration issues when scaling to new settings
>
> We thank the reviewer for their thoughtful criticism. We would like to point out that in practical scenarios, weight $\gamma$ tuning is only necessary to maximize the detection performance of our metric. Therefore, when scaling to new settings one can always start with arbitrary $\gamma$ values to identify some memorized prompts and then later tune these weights to maximize the detection performance. **We have included this in Appendix Section A.5 of our updated paper.**
>
> > The proposed cosine-similarity measure is intuitive but may be sensitive to normalization choices. A sensitivity analysis on score normalization or noise schedule parameters would strengthen the robustness claims.
>
> To demonstrate our method's robustness of our method, **we conduct an ablation study where replace our raw-scores with L1, L2-normalized scores, and spatial L2-normalized scores, along with changing the noise schedule parameters $t$ to multiple early timesteps t={1,2,3}.** The results are presented below.
>
> | Method        | AUC ↑ (SD v1.4) | TPR@1%FPR ↑ (SD v1.4) | AUC ↑ (SD v2.0) | TPR@1%FPR ↑ (SD v2.0) |
> | ------------- | --------------- | --------------------- | --------------- | --------------------- |
> |  |                 |         **Normalization**              |                 |                       |
> | L1            | 0.997           | 0.950                 | 0.937           | 0.804                 |
> | L2            | 0.997           | 0.950                 | 0.937           | 0.804                 |
> | Spatial L2    | 0.995           | 0.934                 | 0.932           | 0.806                 |
> |      |                 |       **Timesteps**                |                 |                       |
> | t=1           | 0.993           | 0.934                 | 0.953           | 0.800                 |
> | t=2           | 0.987           | 0.862                 | 0.933           | 0.794                 |
> | t=3           | 0.985           | 0.860                 | 0.933           | 0.805                 |
>
> The results demonstrate that our approach is robust across all the cases in both SD v1.4 and SD v2.0. **We have included this ablation in Appendix Section A.2.3 of our updated paper.**
>
> > Despite being well-organized, there are several equations in the paper that remain unlabelled on page 5 (between equations 5 and 6).
>
> We thank the reviewer for spotting this. **We have now labelled all the equations in our updated manuscript.**
>
> > The discussion on Figure 2’s overlap could benefit from quantitative metrics (e.g., KL divergence between distributions).
>
> We appreciate the reviewers comment to improve our Figure 2 experiment. **We have now calculated and reported the corresponding KL Divergence values between non-memorized and memorized distributions for anisotropic and isotropic case in our updated manuscript.** Specifically, we find that in isotropy, KDE curves have less overlap with a high KL Divergence (=0.166), indicating better discriminating capabilities of $\|s_\theta^\Delta(\mathbf{x}_t, t ,c)\|$. In contrast, this overlap is higher in anisotropy with a lower KL Divergence (=0.022), thus depicting the failure of norm-based metrics in the low-noise anisotropic case.

---

### Author Response · Authors · 2025-11-22
**Global Response by Authors**

We would like to sincerely thank the reviewers WjxY, y8dV, Kb53 and sxxs for their thorough, insightful and very helpful comments which helped improve our paper immensely. We are glad that the reviewers found our method had following strengths:

* **Clear and sound theoretical framing** linking memorization signatures to isotropy–anisotropy structure in diffusion log-probabilities (**WjxY, y8dV, Kb53**).
* **Novel conceptual shift** from norm magnitude to angular alignment as a more reliable indicator in the low-noise regime (**WjxY, y8dV, Kb53 sxxs**).
* **Efficient, denoising-free metric** requiring only two model evaluations, making it practical for large-scale auditing (**WjxY, y8dV, Kb53**).
* **Comprehensive experiments** with competitive performance across multiple Stable Diffusion models and memorization benchmarks, including new evaluations such as MemBench (**WjxY, Kb53**).
* **Practical significance**, including utility for privacy-preserving generation and effective prompt-level mitigation (**WjxY, y8dV**).

Along with the positive comments, the reviewers also raised very insightful questions and criticisms. In response we have replied to all the comments to reviewers individually. **For the ease of readability, all the changes in the updated paper our written in blue.**

Here, we summarize some of the major changes in our updated manuscript.

* ***Introduced theoretical rigor (sxxs)***- We have now included Theorem 1 (**Section 4.2**) and a proof for the theorem (**Section A.1**), which derives a mathematical bound for cosine similarity through mode distance $\delta$ between $\nabla\_{\mathbf{x\_t}}\log p\_t(\mathbf{x}\_t)$ and $\nabla\_{\mathbf{x\_t}}\log p\_t(c|\mathbf{x}\_t)$
* ***New Ablation Studies (y8dV, WjxY)*** – We expanded our ablations to analyze the contribution and robustness of our metric’s components. Specifically, we (i) evaluated each component individually (**Section A.2.2**), and (ii) examined the effect of different score-normalization schemes and noise-schedule/timestep settings (**Section A.2.3**).
* ***New Experiments on Realistic Vision v5.1 (WjxY)***- To demonstrate the generalizability of our approach, we have now conducted additional detection experiments on one of the latest versions (v5.1) of Realistic Vision model (**Section A.6**).
* ***Explanation on timestep mismatch (y8dV, sxxs)***- We have now provided justification on why probing the model at $t=0$ while inputing gaussian noise at $t=T$ works well in our approach (**Section A.4**).
* ***Provided code and experimental setup for Figure 2 (y8dV, Kb53, sxxs)***- We have uploaded the code as Supplementary Material to this submission and provided experimental details for the FIgure 2 experiment (**Section A.3.1**).
* ***Added comparison with Jeon et al. (2025) (sxxs)***- We have now run the detection metric from Jeon et al. on MemBench and provided the quantitative and qualitative comparisons to our approach (**Section 5.2, Section A.7**)


We once again thank the reviewers and also the Area Chair for their input in our paper.

Sincerely,

Submission #3168 Authors

---

### Author Response · Authors · 2025-11-30

Dear all,

We would first like to extend our sincere appreciation and a very warm welcome to the new area chair assigned to this paper. Understanding the heavy workload assigned to them because of a very unfortunate data leak, we would like to summarize the discussions that were ongoing before the data leak happened.

* The initial scores were 6, 6, 4, and 2.
* In our rebuttal (22.11.25), we addressed all concerns, added new theoretical grounding, and provided multiple new experiments and ablations (detailed in the Global Response and individual replies).
* On 24.11.25 (before the leak became public), **Reviewer Kb53** expressed full satisfaction with our rebuttal (see the Reviewer kb53 thread) and **raised their score from 4 to 8.**
* Other reviewers did not have the opportunity to respond/change scores before communications were halted due to the data leak.

The scores after the update from Reviewer Kb53 were:
* WjxY: 6
* y8dV: 6
* **Kb53: 4--> 8**
* sxxs: 2

We hope this summary makes the task of the newly assigned area chair easier, and we thank both the previous and current area chairs and reviewers for their efforts.

Sincerely,

Submission #3168 Authors

---

### Meta-Review · Area_Chair_qKff · 2026-01-10

**Summary:**

Via both theory and experiments, the paper shows that memorization in diffusion models can be detected and mitigated by calculating a specific metric involving the guidance vector and the unconditional score estimate in the anisotropic low-noise regime.

The scores had high variance and on the somewhat lower side (6,6,4,2) but one reviewer clearly indicated that they would have raised their score (to 8), and the authors seem to have addressed most of the points in the negative review. I think the paper does make natural/useful contributions to the problem of memorization mitigation, and tentatively recommend acceptance.

**Reviewer Concerns:**

See response to the score question below.

**Reviewer Scores:**

Two reviews (scores 6 and 6) would have likely remained unchanged.

One reviewer (score 4) said that they would increase their score. According to the authors, they changed it to 8 before the scores were reverted. I think this is accurate based on the review and response.

One reviewer (score 2) raised concerns about the lack of rigor in the proof of Theorem 1, issues with wording, comparisons (lack thereof) with Jeon '25, and a conceptual question regarding the score computation. I think the first several were addressed in the authors' rebuttal. The last one about score computation/on-manifold correction is somewhat interesting, and I would encourage the authors to think more carefully about that point.

---

### Decision · Program_Chairs · 2026-01-26

Accept (Poster)